JCB | Journal of Cell Biology

# Seipin concentrates distinct neutral lipids via interactions with their acyl chain carboxyl esters

Mike F. Renne[1], Robin A. Corey[2], Joana Veríssimo Ferreira[1], Phillip J. Stansfeld[2,3], and Pedro Carvalho[1]

Lipid droplets (LDs) are essential for cellular lipid homeostasis by storing diverse neutral lipids (NLs), such as triacylglycerol (TAG), steryl esters (SE), and retinyl esters (RE). A proper assembly of TAG-containing LDs at the ER requires Seipin, a conserved protein often mutated in lipodystrophies. Here, we show that the yeast Seipin Sei1 and its partner Ldb16 also promote the storage of other NL in LDs. Importantly, this role of Sei1/Ldb16 is evolutionarily conserved as expression of human-Seipin restored normal SE-containing LDs in yeast Seipin mutants. As in the case of TAG, the formation of SE-containing LDs requires interactions between hydroxyl-residues in human Seipin or yeast Ldb16 with NL carboxyl esters. These findings provide a universal mechanism for Seipin-mediated LD formation and suggest a model for how Seipin distinguishes NLs from aliphatic phospholipid acyl chains in the center of the membrane bilayer.

## Introduction

Lipid droplets (LDs) serve as storage organelles for neutral lipids (NLs), such as triacylglycerol (TAG), sterol esters (SE), and retinyl esters (RE; Olzmann and Carvalho, 2019; Walther et al., 2017). They play a central role in lipid metabolism, and accordingly defects in LD homeostasis are associated with pathologies such as obesity, lipodystrophy, and metabolic syndrome (Herker et al., 2021; Krahmer et al., 2013; Rao and Goodman, 2021).

LDs have a unique structure that consists of an NL core surrounded by a phospholipid (PL) monolayer (Tauchi-Sato et al., 2002). This monolayer architecture is established during LD biogenesis at the ER. The formation of LDs is initiated by NL synthesis. At low concentrations, NLs disperse between the acyl chains of the ER membrane lipids, but at higher concentrations, they coalesce into a lens-like structure (Renne et al., 2020; Thiam and Ikonen, 2021). Growth of this lens leads to the formation of a nascent LD, which buds toward the cytosolic face of the ER giving rise to a mature LD.

LD biogenesis takes place at ER sites marked by Seipin (Choudhary et al., 2020; Grippa et al., 2015; Salo et al., 2016; Wang et al., 2016; Wang et al., 2014; Wang et al., 2018), a conserved ER integral membrane protein. Loss of Seipin leads to the formation of morphologically aberrant LDs, which are either supersized or very small and clustered (Cartwright et al., 2015; Fei et al., 2008; Szymanski et al., 2007; Wang et al., 2014). In addition, the loss of Seipin reduces the rate of LD formation (Cartwright et al., 2015; Grippa et al., 2015) and impairs LD dynamics (Wolinski et al., 2011).

Cryo-EM structures of the human and fly Seipin luminal domain revealed an oligomeric ring-like assembly lined at the inner surface by a hydrophobic helix (HH) that is buried deep into the ER membrane (Sui et al., 2018; Yan et al., 2018). Recent molecular dynamics (MD) simulations showed that the lumenal ring of human Seipin concentrates TAG, even when TAG is present at very low concentrations in the membrane (Prasanna et al., 2021; Zoni et al., 2021). The TAG concentrating activity of human Seipin depends on the HH, requiring two serine residues (S165/166) in this helix (Prasanna et al., 2021; Zoni et al., 2021). Curiously, the membrane protruding HH is absent in the yeast Seipin Sei1 (Klug et al., 2021; Arlt et al., 2022), and consequently, Sei1 alone does not appear to enrich TAG (Klug et al., 2021). Instead, Sei1 is complemented by Ldb16, a yeast-specific Seipin binding partner (Grippa et al., 2015; Wang et al., 2014), which concentrates TAG via a short helix rich in hydroxyl-containing residues (Klug et al., 2021).

The number, size, and composition of LDs vary greatly depending on cell type and metabolic state (Olzmann and Carvalho, 2019; Thiam and Beller, 2017). In certain cells, such as adipocytes and hepatocytes, LDs mainly contain TAG. In contrast, foam cells, lipid-laden macrophages found in atherosclerotic lesions, contain LDs that mainly consist of SE (Stary et al., 1994). In addition, other structurally diverse NLs are stored in LDs, such as RE in hepatic stellate cells, and acylceramides were found in both yeast and mammals (Thiam and Ikonen, 2021). Despite the diversity of NLs stored in LDs, the process of LD biogenesis has been studied almost exclusively for

[1]Sir William Dunn School of Pathology, University of Oxford, Oxford, UK;   [2]Department of Biochemistry, University of Oxford, Oxford, UK;   [3]School of Life Sciences and Department of Chemistry, University of Warwick, Coventry, UK.

Correspondence to Pedro Carvalho: pedro.carvalho@path.ox.ac.uk.

**Rockefeller University Press**
J. Cell Biol. 2022 Vol. 221 No. 9   e202112068



TAG-containing LDs. Whether the current model of LD biogenesis also applies to other NLs remains unknown.

Here, we compared the biogenesis of LDs containing exclusively TAG or SE. We found that the yeast Seipin Sei1 and its binding partner Ldb16 are required for the formation of both types of LDs. Importantly, defects in TAG- or SE-containing LDs in yeast Seipin mutants were rescued by expression of human Seipin, indicating a conserved role for Seipin in the formation of TAG- and SE-LDs. MD simulations and mutagenesis studies show that human Seipin efficiently concentrated TAG, SE, as well as other NLs using a similar mechanism. This activity involved interactions between NLs and the HH, likely via hydrogen bonds (H-bonds) between conserved hydroxyl-containing amino acids and carboxyl esters present in TAG and SE. Taken together, our data indicate that Seipin has a general role in the packaging of structurally distinct NLs by direct interactions with their acyl chain carboxyl esters.

## Results

### Sei1 and Ldb16 localization is independent of neutral lipids

In yeast (*Saccharomyces cerevisiae*), the synthesis of TAG and SE depends on the diacylglycerol acyltransferases (DGAT) Dga1/Lro1 and the sterol *O*-acyltranferases (SAT) Are1/Are2, respectively (Oelkers et al., 2000, 2002; Sorger and Daum, 2002; Yang et al., 1996; Yu et al., 1996). Although NL synthesis is essential in higher eukaryotes and other yeasts (Zhang et al., 2003), a budding yeast quadruple mutant lacking all four acyltransferases (*dga1Δlro1Δare1Δare2Δ*, hereafter called no NL cells) is viable. As this mutant lacks NLs, it is consequently devoid of LDs (Petschnigg et al., 2009; Sandager et al., 2002). In cells devoid of NLs, synthesis of either TAG or SE by the expression of a single DGAT or SAT is sufficient to induce LD formation (Sandager et al., 2002), indicating that yeast can produce LDs consisting of only TAG or SE. To compare the mechanisms by which TAG and SE are packaged into LDs, we utilized yeast mutants that synthesize only one type of NL, either TAG-only (*are1Δare2Δ*) or SE-only (*dga1Δlro1Δ*; Fig. S1 A).

WT, TAG-only, and SE-only cells were cultured in synthetic defined media devoid of inositol, a condition widely used to stimulate the biosynthesis of NLs and formation of LDs (Henry et al., 2012; Fei et al., 2011). LDs from the various cells were stained with the neutral lipid dye BODIPY and imaged by super-resolution spinning disc confocal microscopy (Fig. 1 A and Fig. S1 B). The BODIPY signal was used to quantify LDs as described in Fig. S1 C. In relation to WT cells, TAG-only cells displayed a lower number of LDs, but these were slightly bigger (Fig. 1, A and B). In contrast, in SE-only cells, LDs were much smaller and less abundant, with a population of SE-only cells lacking LDs (Fig. 1, A and B). These observations are consistent with lower amounts of SE in comparison to TAG (Casanovas et al., 2015; Connerth et al., 2010; Hariri et al., 2018; Sandager et al., 2002; Zweytick et al., 2000). In cells with no NLs, BODIPY partitions to other hydrophobic structures such as lipid bilayers resulting in dim labeling of all cellular membranes (Fig. 1 A), as described previously (Wolinski and Kohlwein, 2015).

We next analyzed the localization of key proteins involved in LD formation in TAG-only and SE-only cells. In WT cells, Sei1 and Ldb16 display punctate distribution in the ER as previously described (Choudhary et al., 2020; Wang et al., 2014). A similar distribution was also observed in TAG-only and SE-cells (Fig. 1, C and D). Importantly, LDs were adjacent to both Sei1 and Ldb16 foci, irrespective of the LD NL content, consistent with the role of these proteins in LD biogenesis and stabilization of ER-LD contacts (Grippa et al., 2015; Salo et al., 2016). In addition, the localization of proteins involved in LD budding (Scs3/Yft2 and Pex30; Choudhary et al., 2015; Joshi et al., 2018; Kadereit et al., 2008; Wang et al., 2018) and DAG synthesis (Nem1; Karanasios et al., 2010) was indistinguishable between WT, TAG-only, and SE-only cells as well (Fig. S1 D). Curiously, the LD-coating perilipin Pln1, previously reported to bind specifically TAG-containing LDs (Gao et al., 2017), also showed robust localization to SE-only LDs (Fig. 1 E). However, consistent with earlier findings (Gao et al., 2017), Pln1 levels were lower in SE-only cells and were virtually undetectable in no LD cells both by microscopy (Fig. 1 E) and Western blotting (Fig. S1 E). This is in agreement with the observation that the levels of Pln1 and other perilipins are dependent on LD abundance (Gao et al., 2017; Mishra et al., 2016).

Thus, the localization of key factors involved in LD formation appears consistent between LDs formed from different NLs.

### Temperature-induced lipid changes in SE-only cells result in increased LDs

Under standard culture conditions (i.e. cultured at 30°C), SE-only cells had only few small LDs (Fig. 1, A and B), complicating the analysis of LD-related phenotypes. Serendipitously, we observed that when cultured at elevated temperature (37°C), SE-only cells had increased numbers of LDs (>65% having three or more LDs at 37°C vs. <30% at 30°C), that were also slightly bigger (0.52 ± 0.18 nm at 37°C vs. 0.46 ± 0.17 nm at 30°C; Fig. 2, A and B; and Fig. S2 A). Interestingly, this effect appeared specific to SE-only LDs as TAG-only cells had slightly smaller LDs, while LD size in WT cells was comparable between cells grown at 30 and 37°C (Fig. 2, A and B).

To gain insights into these temperature-dependent changes in LDs, we compared the lipidomes of WT, TAG-only, and SE-only cells grown to stationary phase at 30 and 37°C. We observed little changes in the lipidomes of WT cells grown at 30 and 37°C (Fig. 2, C and D; and Fig. S2 B), with the exception of increased saturation of phospholipid acyl chains (Fig. 2 D) as expected due to the homeoviscuos adaptation response (Klose et al., 2012; Renne and de Kroon, 2018). The lipid composition of TAG-only cells was very similar to WT cells at both temperatures (Fig. 2 C and Fig. S2 B). In contrast, SE-only cells displayed striking changes in their lipidome when compared to WT and TAG-only cells (Fig. 2 C). In SE-only cells, the loss of TAG resulted in a concomitant increase in the levels of the TAG precursor DAG as well as phospholipids, more prominently phosphatidylcholine and phosphatidic acid (Fig. 2 C). These cells showed a higher fraction of di-unsaturated phospholipids (Fig. 2 D). Moreover, SE-only cells showed a robust increase in SE levels, particularly at 37°C. These results are consistent with the increased number

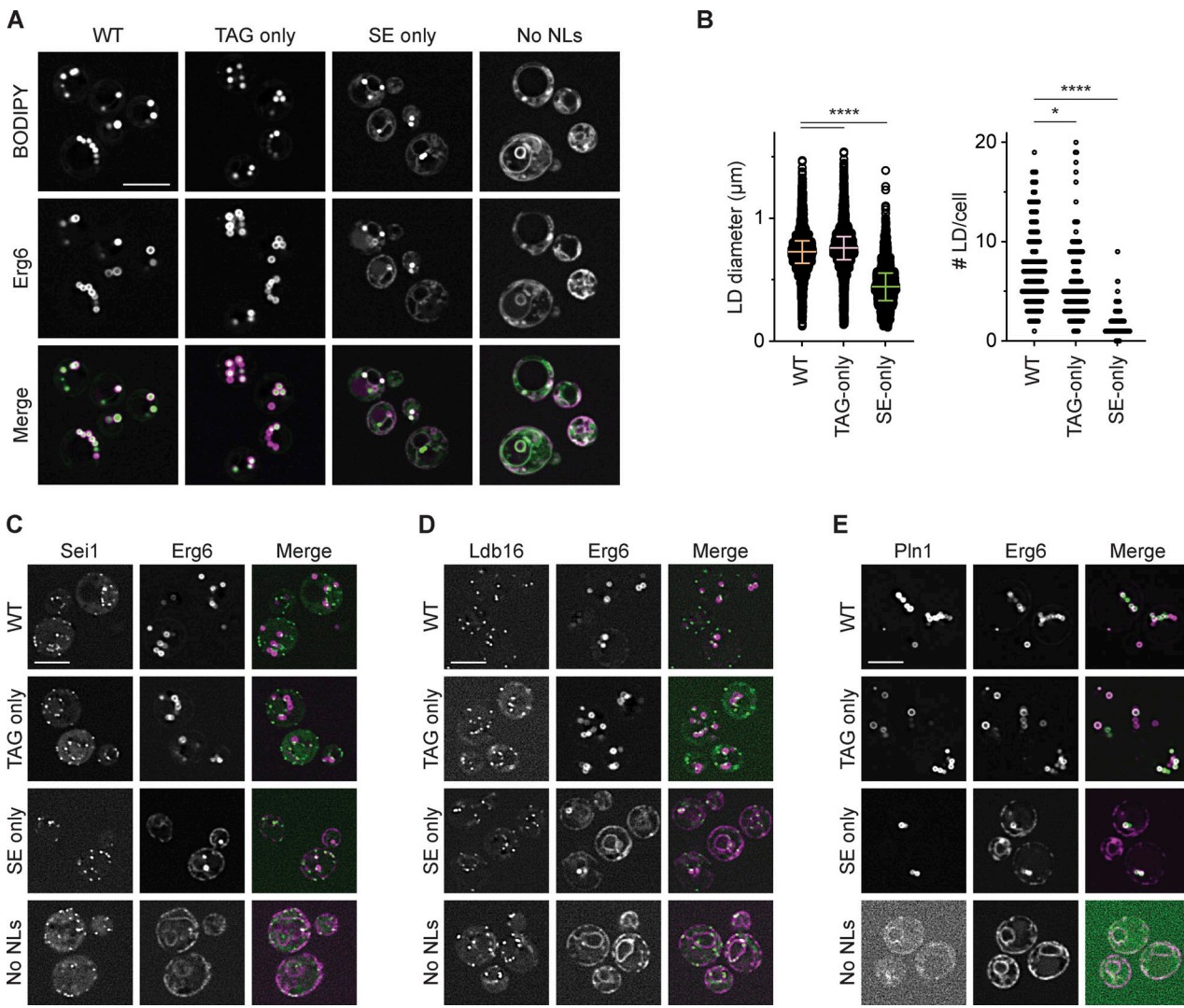

Figure 1. **Characterization of lipid droplets in WT, TAG-only, SE-only cells. (A)** Fluorescence micrographs of LDs in WT, *are1Δare2Δ* (TAG-only), *dga1Δlro1Δ* (SE-only), and *dga1Δlro1Δare1Δare2Δ* (no NLs). Cells were cultured to late stationary phase, and LDs were visualized by BODIPY-staining and the LD marker protein Erg6-mCherry. Scale bars correspond to 5 μm. **(B)** Quantification of LD size and LD per cell in cells with the indicated genotype (labeled according to A). LD size was quantified for a minimum of 150 LDs/genotype and at least 100 cells/genotype were quantified to determine the number of LD per cell. Difference in LD size and LD/cell was analyzed by non-parametric testing (Kruskal–Wallis test) followed by Dunn's multiple comparison analysis (*, P < 0.05; ****, P < 0.0001). **(C–E)** Localization of Sei1 (C), Ldb16 (D) and Pln1 (E) in WT, TAG-only, SE-only, or no LDs cells. Proteins of interest were expressed from their endogenous loci as C-terminal fusions to mNeonGreen and Erg6-mCherry was used as LD marker. Scale bars correspond to 5 μm.

and size of LDs in SE-only cells grown at 37°C, likely due to higher SE levels and changes in ER phospholipid composition.

Thus, the temperature-dependent changes in LD number and size provide a simple and robust platform to study the role of Seipin in the formation of SE-only LDs.

## Packaging of structurally diverse NLs into LDs requires Sei1 and Ldb16

We next investigated the requirement of Sei1 and Ldb16 in the formation of LDs containing different NLs by analyzing cells grown at 30 and 37°C. In WT or TAG-only cells, loss of Sei1, Ldb16, or both resulted in defects LD morphology (Fig. 3, A and B), as expected (Cartwright et al., 2015; Wang et al., 2014). In these cells, the normal (or Gaussian) distribution of LD sizes was lost in Seipin mutants, which accumulated populations of tiny as

well as supersized LDs (Fig. 3 B). The defects in LD morphology had a comparable magnitude between cells grown at 30 and 37°C. Importantly, the loss of Seipin complex in SE-only cells also resulted in significant defects in LD morphology. While these defects were small in cells grown at 30°C (Fig. 3 A and Fig. S1 C), they were pronounced in SE-only cells grown at 37°C upon loss of *sei1Δ*, *ldb16Δ*, and *sei1Δ ldb16Δ*, which accumulated both supersized and tiny LD populations (Fig. 3, C and D).

LDs predominantly made of SEs are also found in cells lacking the phosphatase Nem1. Nem1 activates the phosphatidic acid hydrolase Pah1, the main enzyme that produces DAG for TAG synthesis (Hsieh et al., 2016; Pascual et al., 2013). In *nem1Δ* cells, TAG levels are markedly reduced, leading to low numbers of LDs consisting mainly of SE (Adeyo et al., 2011; Pascual et al., 2013). Loss of Sei1, Ldb16, or both in the *nem1Δ* background resulted in

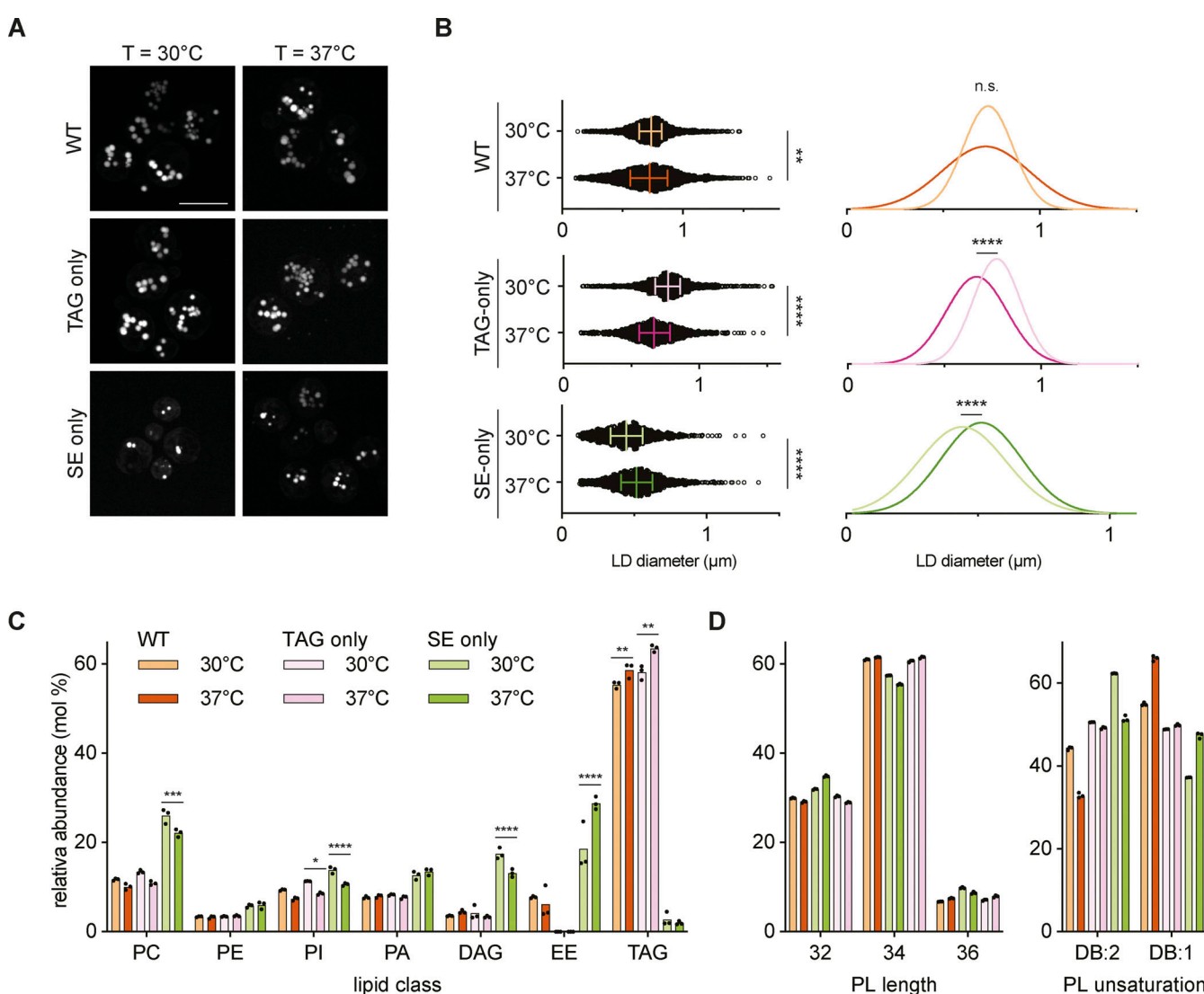

Figure 2. **Cultivation at elevated temperature increases LD number and size in SE-only cells. (A)** Fluorescence micrographs of LDs in WT, TAG-only, and SE-only cells cultured in inositol-free media to late stationary phase at the indicated temperature. LDs were stained with BODIPY, scale bars correspond to 5 µm. **(B)** Quantification of LD size in WT, TAG-only, and SE-only cells cultured at indicated temperatures as in A. Left: plot of individual LD diameter measurements. Bars and whiskers indicate mean with interquartile range. LDs were quantified in $n$ = 3 biological independent experiments, with a minimum of 1,100 total LDs counted per genotype. Difference in distribution of LD size was analyzed by non-parametric testing (Kruskal–Wallis test) followed by Dunn's multiple comparison analysis (**, P < 0.01; ****, P < 0.0001). Right: Non-linear regression fitted Gaussian-curves of histograms of LD diameter frequency distribution (bin-width 0.05 µm). Difference in mean of the Gaussian fits was compared by extra sum-of-squares F-test (****, P < 0.0001; n.s., non significant). **(C)** Lipid class composition in mol% of total lipids analyzed. Lipids classes detected <2% are shown (phosphatidylcholine, PC; phosphatidylethanolamine, PE; phosphatidylinositol, PI; phosphatidic acid, PA; diacylglycerol, DAG; ergosteryl ester, EE; triacylglyerol, TAG). Bars indicate mean with individual datapoints ($n$ = 3) shown. Indicated mutants cultured to stationary phase at $T$ = 30°C (light bars) or $T$ = 37°C (dark bars). EE was not detected in the TAG-only mutant. Differences were tested using two-way ANOVA followed by multiple comparisons with Tukey correction (**, P < 0.01; ***, P < 0.001; ****, P < 0.0001). **(D)** Glycerophospholipid (GPL) total acyl chain unsaturation (left) and length (right) in mol% of total GPL. Bars indicate mean with individual datapoints ($n$ = 3) shown.

LD morphology defects similar to the ones observed in SE-only cells (Fig. S2 C), further suggesting that Seipin promotes the formation of LDs enriched in SEs.

We next tested whether loss of Seipin affected the rate of SE-only LD formation. Expression of the SE acyltransferase *ARE1* in no NLs cells resulted in time-dependent LD formation with ~75% of cells displaying one or more LDs after 60 min of *ARE1* induction (Fig. 3 E). Both *sei1Δ* and *ldb16Δ* mutants showed a pronounced delay in the formation of SE-only LDs, with a reduction in the number of LDs at all timepoints analyzed (Fig. 3

E). Moreover, the LDs formed in the mutant cells displayed abnormal morphology that became even more apparent upon long-term overexpression (24 h) of Are1 as well as its homolog Are2 (Fig. S2 D). Thus, as observed for TAG LDs (Cartwright et al., 2015; Grippa et al., 2015; Fig. S2 E), the Sei1/Ldb16 Seipin complex is required for the normal formation of SE-containing LDs.

Recently, the expression of human lechitin retinol acyltransferase (LRAT) in no NLs cells was shown to lead to the formation of retinyl ester (RE)-containing LDs (Molenaar et al.,

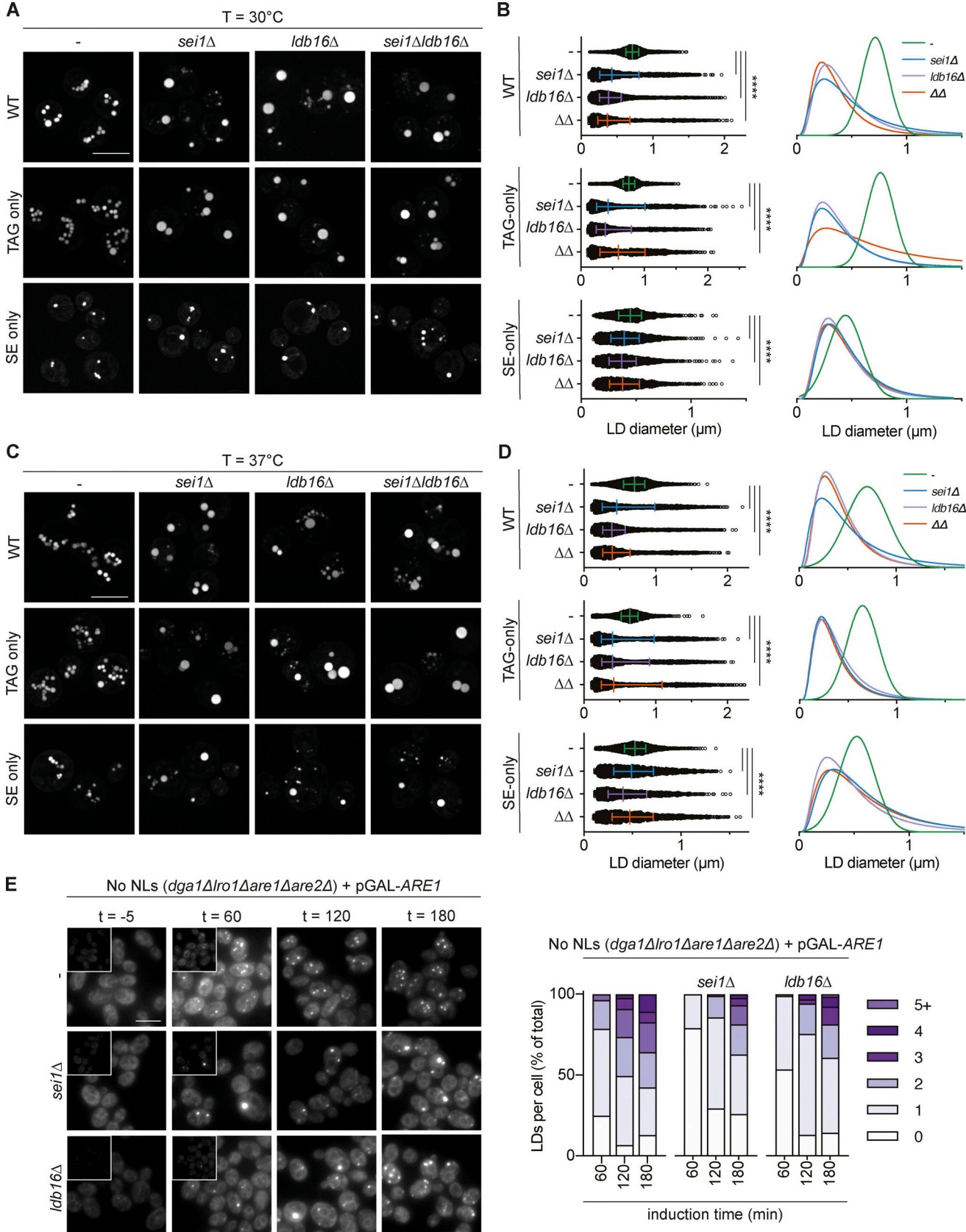

**Figure 3.** **Sei1/Ldb16 Seipin complex is required for normal morphology and biogenesis of TAG-only and SE-only LDs. (A)** Fluorescence micrographs of LDs in cells with indicated genotypes. Cells were cultured in inositol-free media at 30°C to late stationary phase, and LDs were stained with BODIPY. Scale bars correspond to 5 µm. **(B)** Quantification of LD size in cells with indicated genotype grown as in A. Left: of individual LD diameter measurements. Bars and

whiskers indicate mean with interquartile range. LDs were quantified in *n* = 4 biological independent experiments (*n* = 3 for *sei1Δldb16Δ; ΔΔ*), with a minimum of 1,500 total LDs counted per genotype. The difference in distribution of LD size was analyzed by non-parametric testing (Kruskal–Wallis test) followed by Dunn's multiple comparison analysis (****, P < 0.0001). Right: non-linear regression fitted curves of histograms of LD diameter frequency distribution (bins 0.05 μm). Non-linear regression tested fits of normal (Gaussian) or log-normal curves and the most probable fit (>99.99% certainty) is shown. **(C)** As in A, but cells were grown at 37°C. **(D)** As in B, but cells were grown at 37°C. **(E)** Analysis of LD biogenesis upon induction of SE synthesis by expression of the SE acyltransferase Are1. Plasmid-borne expression of Are1 was induced by the addition of galactose (final concentration 2%) to cells with the indicated genotype grown in raffinose media. LDs were stained with BODIPY at indicated time points (in minutes after galactose induction). Scale bars correspond to 5 μm. The number of LDs per cell was counted in a minimum of 50 cells per genotype per timepoint.

2021). We tested whether the packaging of RE, an NL naturally not present in yeast, also required Sei1 and Ldb16. Induction of LRAT in no LD cells supplemented with retinol resulted in the formation of LDs visualized by the LD marker protein Erg6 (Fig. S2 F). Induction of RE synthesis in *sei1Δ* or *ldb16Δ* resulted in the formation of larger LDs, indicating a defect in RE LD formation (Fig. S2, F and G).

Collectively, these results indicate that the yeast Sei1/Ldb16 Seipin complex is required for the correct formation of LDs irrespective of their NL content.

## Human Seipin binds NLs via hydrogen bonding to carboxyl esters

Expression of human Seipin rescues LD defects of *sei1Δ*, *ldb16Δ*, and *sei1Δldb16Δ* cells (Wang et al., 2014). Taking advantage of this observation and MD simulations showing TAG concentrating activity for human Seipin (Prasanna et al., 2021; Zoni et al., 2021), we asked whether human Seipin could also concentrate other NLs.

We performed coarse-grained MD simulations of human Seipin in a complex ER bilayer (see Materials and methods) with 1% TAG and 3% SE. After 5 μs, there was a strong enrichment of both NLs within the Seipin ring (Fig. 4, A and B), while most membrane lipids were depleted (Fig. 4 B). Enrichment of SE was also observed when it was the only NL in the system (Fig. S3 A), as observed for TAG (Prasanna et al., 2021; Zoni et al., 2021; Fig. S3 B). Importantly, at the concentrations used in our simulations, the frequency of NL nucleation is low in the absence of Seipin (Fig. S3 C). Thus, the enrichment likely reflects direct interactions of the various NLs with Seipin.

Next, we investigated in further detail the interactions of Seipin luminal domain with TAG and SE. Both NLs were found in close proximity to the HH that protrudes into the ER bilayer (Fig. 4 C), with S165/166 and adjacent hydrophobic amino acids (L159, L162, and V163) as the main interacting residues (Fig. 4 D; and Fig. S3, A and B). Similar results were observed when this analysis was performed with RE (Fig. S3 D), consistent with in vivo experiments in yeast (Fig. S2, E and F). Thus, SE and RE interact with the same sites as TAG (Prasanna et al., 2021; Zoni et al., 2021), strongly suggesting a general mechanism for Seipin–NL interactions.

Seipin S165/166 residues are critical for concentrating TAG, likely through the formation of H-bonds (Prasanna et al., 2021; Zoni et al., 2021). As we observed that the same residues are involved in the interactions with SE and RE, we searched for the NL chemical groups that preferentially interact with Seipin S165/166 residues. Lipid occupancy analysis, which reports on the number of simulation frames that each residue is within

0.6 nm of a given lipid or lipid bead, showed that S165/166 predominantly interacts with TAG glycerol backbone and the three carboxyl esters (Fig. 5 A). In the case of SE, S165/166 predominantly interacted with the single carboxyl ester (Fig. 5 B), and occupancy of the acyl chain and sterol backbone decreases depending on the distance from the carboxyl group. These results indicate that Seipin Ser165/166 interacts with TAG and SE-based carboxyl esters as a common functional group.

Similar to PLs, free fatty acids (FFA) were not enriched by Seipin and were excluded from the Seipin ring, either in the presence or absence of TAG (Fig. S3 E). On the other hand, squalene (SQL), an isoprenoid lipid lacking carboxyl esters, was enriched within the Seipin ring (Fig. S3 F). However, consistent with the absence of carboxyl esters, the binding duration of SQL with Seipin S165/166 was greatly diminished when compared to TAG, SE, or RE (Fig. S3 G). The molecular basis for the observed SQL concentration is unclear, but curiously, in vivo SQL is only efficiently stored in LDs in the presence of TAG and SE, and in cells devoid of these NLs it accumulates at high concentrations in the ER bilayer (Spanova et al., 2010).

As carboxyl esters are also present in membrane PLs, we wondered how Seipin binds specifically to NLs. Density analysis along the bilayer perpendicular revealed that S165/166 are positioned deep into the ER membrane, close to the membrane equator (Fig. S3 H). As NLs disperse amongst the PL acyl chains, their carboxyl esters are abundant in this area. In contrast, the density for PL carboxyl esters is more peripheral, proximal to membrane interfaces (Fig. S3 H). Thus, insertion depth of the HH likely precludes interactions with PL carboxyl esters and confers specificity to NLs.

## Hydroxyl-residues in human Seipin and yeast Ldb16 are required for efficient packaging of TAG and SE into LDs

To assess the role of Seipin's NL-interacting S165/166 residues in vivo, we exploited the observation that the expression of human Seipin restored normal LD morphology in *sei1Δldb16Δ* mutants (Wang et al., 2014; Fig. 6, A and B; and Fig. S4, A and B). Expression of human Seipin also restored normal morphology of TAG-only and SE-only LDs (Fig. 6, A and B; and Fig. S4, A and B). Thus, when expressed in yeast, human Seipin promotes normal LD morphology, irrespective of NL content. Mutation of S165/S166 to alanine (SS-AA) did not impair LD morphology (Fig. 6, A and B), possibly due to high expression levels driven by the strong *ADH1* promoter. Importantly, mutation of Seipin S165/166 to aspartate (SS-DD) failed to restore normal LD morphology in *sei1Δldb16Δ* cells (Fig. 6, A and B; and Fig. S4, A and B), even if the expression levels of these mutant proteins were comparable to

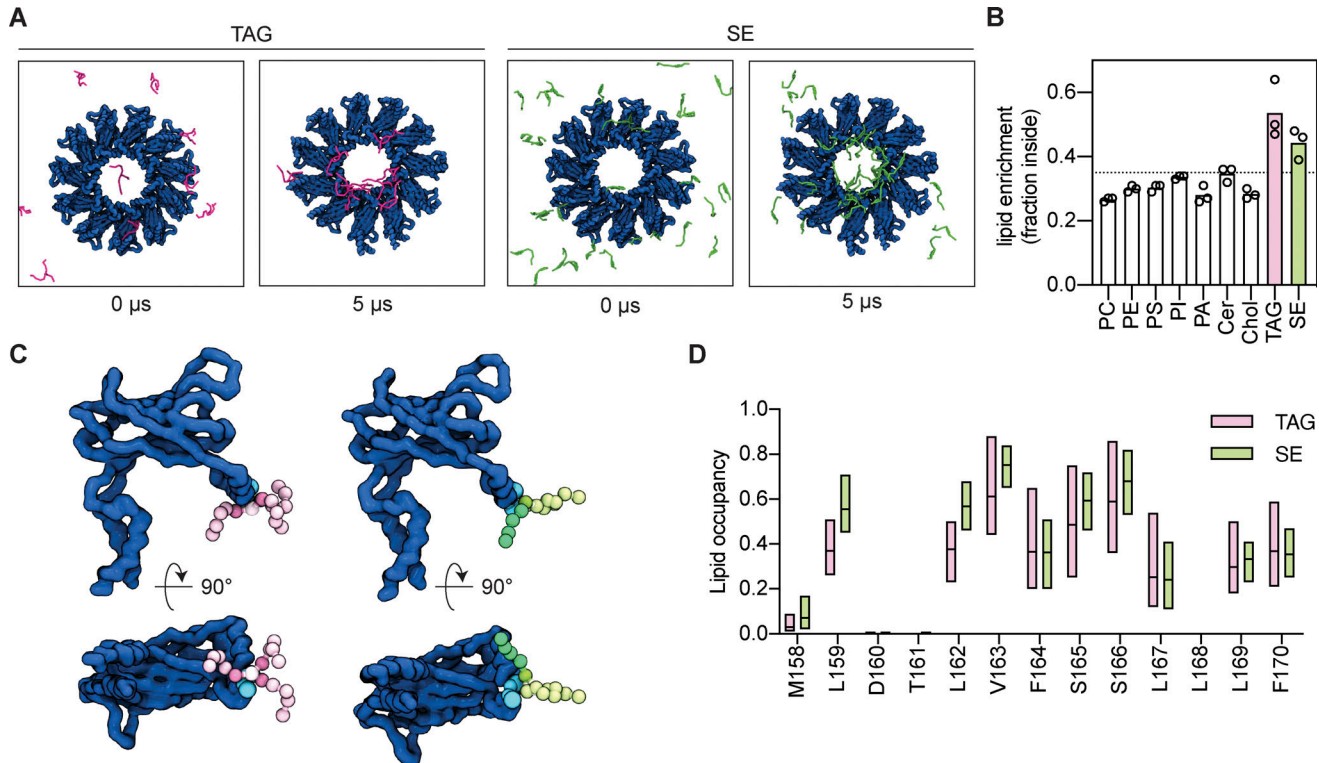

**Figure 4. Seipin enriches TAG and SE via interactions with its hydrophobic helix. (A)** Initial (0 µs) and final (5 µs) top-view snapshots of coarse-grained MD simulations showing enrichment of TAG (pink, left panels) and SE (green, right panels) by human Seipin (blue). **(B)** Analysis of lipid enrichment by human Seipin shown as a fraction of total lipids inside the Seipin ring. The dashed line indicates the fraction of the membrane surface occupied by the Seipin ring. Bars represent the average of three independent simulations (circles). **(C)** Snapshot of MD simulations highlighting human Seipin interaction with TAG (left, pink) and SE (right, green). Human Seipin is shown in dark blue, with S165/166 in the HH depicted as light blue spheres. NL beads are colored as shown in panel E/F, with the carboxyl esters in dark pink (TAG) or dark green (SE). Poses were generated using *PyLipID* (Song et al., 2022). **(D)** Analysis of human Seipin HH residues occupancy by TAG (pink) and SE (green). Bars indicate minimum to maximum values with a line at the median.

WT human Seipin (Fig. S4 C). Consistently, the Seipin SS-DD mutants also showed lower rates of LD formation upon induction of SE production (Fig. S4 D) and accumulated aberrant LDs upon extended overexpression of Dga1 or Are1 (Fig. S4 E). Along with the in silico experiments, these data strongly indicate that human Seipin uses S166/165 to concentrate both TAG and SE during LD formation.

We recently showed that TAG concentration by the yeast Seipin complex is mediated by Ldb16 via a short helix that is rich in hydroxyl residues (T52/S53/S55/T61/S62/T63). Mutation of these hydroxyl residues to alanine (6xA) impairs LD morphology, without affecting protein expression levels (Klug et al., 2021). Importantly, LD morphology defects were observed in TAG-only and SE-only cells expressing Ldb16-6xA (Fig. 7, A and B; and Fig. S5, A and B). In addition, cells expressing Ldb16-6xA displayed lower rates of SE-only LD formation (Fig. S5 C) and LD morphology defects upon extended overexpression of Dga1 or Are1 (Fig. S5 D).

Taken together, these experiments show that both yeast and human Seipin complexes depend on hydroxyl-containing residues to promote the biogenesis of LDs containing distinct NL classes, indicating they utilize a similar mechanism to enrich NLs.

## Discussion

Seipin is central to LD biogenesis, yet its molecular function remains elusive. Recent studies employing imaging, structural, and MD simulation approaches proposed a mechanism by which the Seipin concentrates TAG in its ring-like structure for LD formation (Sui et al., 2018; Yan et al., 2018; Klug et al., 2021; Prasanna et al., 2021; Zoni et al., 2021; Arlt et al., 2022). In contrast, the role of Seipin in the packaging of other structurally distinct NLs into LDs still is obscure, and there are suggestions for Seipin-independent LD assembly (Molenaar et al., 2021; Sołtysik et al., 2021). Here, we show that besides TAG, Seipin promotes the assembly of LDs containing other NLs, such as SE and RE, supporting a universal role for Seipin in NL storage into LDs.

Mutations in yeast Seipin Sei1 or its partner Ldb16 showed defects in LD formation and morphology in all conditions tested, including different temperatures and expression levels of NL acyl transferases. While defects in TAG-only LDs were strong under all conditions, the defects in SE-only LDs became stronger in cells grown at 37°C or upon SAT overexpression (Fig. 3, C and D; and Fig. S2 C). The increased SE levels and consequent increase in LD numbers under these conditions (Fig. 2 C) likely contributed to make the role of Seipin in SE LD formation more

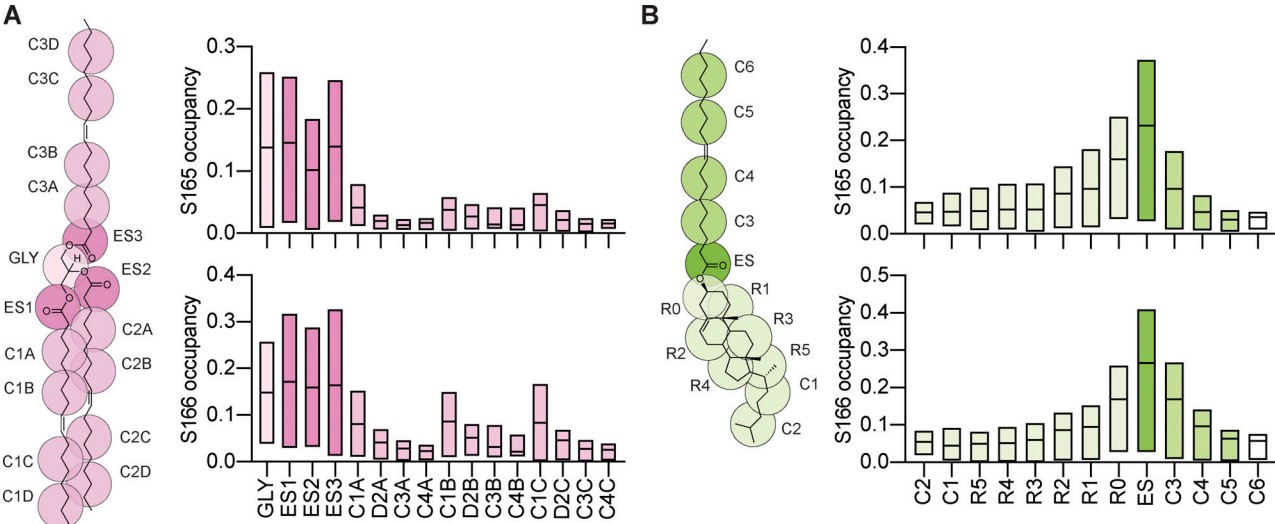

**Figure 5.** **Seipin Serine165/166 interacts with NL carboxyl esters.** Analysis of NL chemical group occupancy by Seipin S165 (top) and S166 (bottom). **(A and B)** Coarse-grained representations of the lipid beads of trioleoylglycerol (TAG, A) and cholesteryl oleate (SE, B) are shown on the respective molecular structures. Bars indicate minimum to maximum values with a line at the median.

apparent. Besides the higher SE levels, SE-only cells grown at 37°C had additional changes in their lipidome. In particular, the reduction in DAG and the major phospholipids PC and PI, as well as the increase in acyl chain saturation have been shown to affect the efficiency of LD formation (Ben M'barek et al., 2017; Zoni et al., 2021a; Adeyo et al., 2011; Wang et al., 2018) and likely contributed to the increase of SE LDs in cells grown at 37°C.

Mutations in Seipin also resulted in aberrant RE-only LDs triggered by overexpression of LRAT (Fig. S2 G). Our results conflict with recent observations by Molenaar and colleagues (Molenaar et al., 2021), that suggest that RE and SE LDs form independently of Seipin. Differences in growth conditions and/or protein expression levels, which impact both NL levels and membrane lipid composition, may explain these seemingly conflicting results.

Conserved hydroxyl-containing residues in the lumenal helix of human Seipin or equivalent residues in yeast Ldb16 concentrate structurally diverse NLs. In all cases, the hydroxyl-containing residues interact likely via a H-bond with NL carboxyl ester groups. Since most NLs contain esterified acyl chains (Thiam and Ikonen, 2021), this supports a general mechanism for Seipin binding to structurally diverse NLs (Fig. 8). While supported by in silico and mutagenesis experiments, the direct interaction between Seipin hydroxyl-containing residues and NLs carboxyl esters should be explored in future studies. These should also investigate whether Seipin's transmembrane segments contribute to binding to SE and other NLs, as was recently suggested for DAG and TAG (Klug et al., 2021; Zoni et al., 2021).

Many NLs and all PLs and FFAs contain carboxyl ester or carboxyl groups that could in principle form H-bonds with Seipin hydroxyl residues. However, the projection of Seipin HH deep into the membrane bilayer positions S165/166 close to the NL carboxyl esters at the center of the bilayer and distant from the PL carboxyl esters and FFA carboxyls, which are at the

membrane–water interface (Fig. S2 G and Fig. 4 E). Thus, the position of Seipin HH is critical for its preferred interaction with NLs over PLs. This position also seems sufficient to recruit NLs without H-bonding groups, such as SQL, albeit with a lower interaction with the S165/166 residues (Fig. S2 F). This is possibly due to Seipin allowing a more favorable geometry for these lipids than when free in the bilayer, although additional work will be needed to assess this.

It should be noted that, whilst the MARTINI 2.2 force field does implicitly model H-bonds, the lack of explicit description is a limitation (Jarin et al., 2021; Alessandri et al., 2019). Future work on the nature of Seipin–NL interactions may benefit from the use of the MARTINI 3 force field (Souza et al., 2021), which includes a more accurate description of H-bonds but, at the time of publication, was still without the full complement of NLs needed for this study. Additionally, the use of fully atomistic simulations may be of benefit for a more detailed understanding of these interactions.

Besides Seipin, other LD regulatory factors may use a similar H-bonding mechanism to specifically recognize NLs in the hydrophobic membrane environment. For example, targeting of the mammalian CTP:phosphocholine cytidylyltransferase enzyme (CCTα) to LDs was shown to be facilitated by a direct interaction between NLs and a critical tryptophan residue on its amphipathic helix, likely via a H-bond (Kim et al., 2021). Thus, this mode of protein–NL interaction may play a general role in LD biology.

Our data suggest that Seipin is an LD assembly factor of broad NL-specificity. Positioning of Seipin hydroxyl residues at the bilayer normal allows Seipin to interact with a diverse range of NLs based on the presence of acyl chain carboxyl esters as a general NL property. This model allows Seipin to enrich structurally distinct NLs across many cell types and organisms and may explain its ubiquitous expression and high conservation.

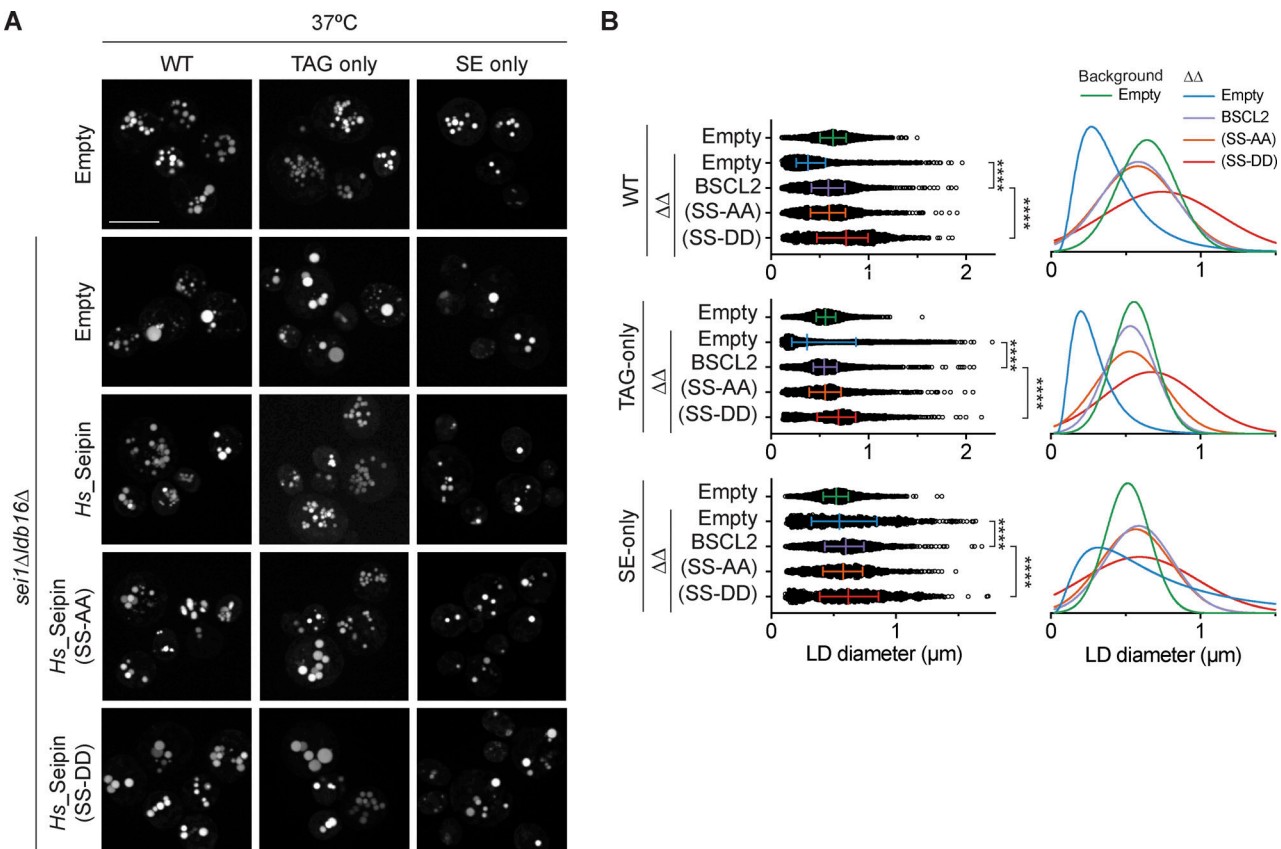

**Figure 6. Hydroxyl-containing residues in human Seipin are required for correct LD morphology in TAG-only and SE-only cells. (A)** Analysis of LD morphology in cells with the indicated genotype by fluorescence microscopy. Human Seipin (*Hs*_Seipin) or indicated mutants were expressed in *sei1Δldb16Δ* (*ΔΔ*) mutants in indicated background strains. Cells were cultured in inositol-free media at 37°C to late stationary phase, and LDs were stained with BODIPY. Scale bars correspond to 5 μm. **(B)** Quantification of LD size in cells with indicated genotype grown as in A. Left: of individual LD diameter measurements. Difference in distribution of LD size was analyzed by non-parametric testing (Kruskal–Wallis test) followed by Dunn's multiple comparison analysis (**, P < 0.01, ****, P < 0.0001). Right: non-linear regression fitted curves of histograms of LD diameter frequency distribution (bins 0.05 μm). Non-linear regression tested fits of normal (Gaussian) or Log-normal curves and most probable fit (>99.99% certainty) is shown.

In most cells, the LD core consists of multiple NL classes and species (Thiam and Beller, 2017). Interactions between different NLs likely occur already during the LD biogenesis process in the ER bilayer, potentially influencing the behavior of individual NLs and their partitioning into LDs. Our work takes advantage of cells engineered to produce a single class of NLs as a simplified system to study the role of Seipin in the packaging of individual NLs. Similar reductionist approaches have been important in revealing the contributions of individual proteins, lipids, and membrane biophysical properties during LD formation (Ben M'barek et al., 2017; Chorlay and Thiam, 2018; Chorlay et al., 2019; Zoni et al., 2021a; Choudhary et al., 2018). In the future, it will be interesting to dissect the interplay between multiple NLs during LD formation. Interestingly, certain cell types can store specific NLs, such as adipocytes, foam cells, and hepatic stellate cells that contain mainly TAG, SE, and RE, respectively (Thiam and Ikonen, 2021; Thiam and Beller, 2017). Furthermore, different LD populations enriched in specific NLs have been identified within a single cell type (Hsieh et al., 2012; Di Napoli et al., 2016). The conditions used in our study may mimic LD formation in these exceptional cases.

# Materials and methods
## Yeast strains and plasmids
Yeast strains used in this study were derived from BY4741, BY4742, or BY4743 (Brachmann et al., 1998), and are listed in Table S1. Plasmids used are based on pRS316, pRS415, or pRS416 (Sikorski and Hieter, 1989; Christianson et al., 1992) and are listed in Table S2.

## Media and culture conditions
All strains were cultured in synthetic media at 30 or 37°C as indicated while shaking at 200 rpm. Strains were pre-cultured overnight, inoculated at OD 0.1, and cultured for 24 h to late stationary phase. Synthetic defined glucose media devoid of inositol (SC I-) contained per liter the following: 6.7 g yeast nitrogen base without inositol (MP Biomedicals), 0.6 g complete supplement mix without his/leu/met/ura (MP Biomedicals), 20 g D-glucose (Sigma-Aldrich) and was supplemented with amino acids as required.

For induction of NL acyltransferases, cells were precultured in synthetic defined raffinose media and galactose was added to a final concentration of 2% (wt/vol from a 20% stock). Synthetic defined raffinose media contained the following per liter: 6.7 g

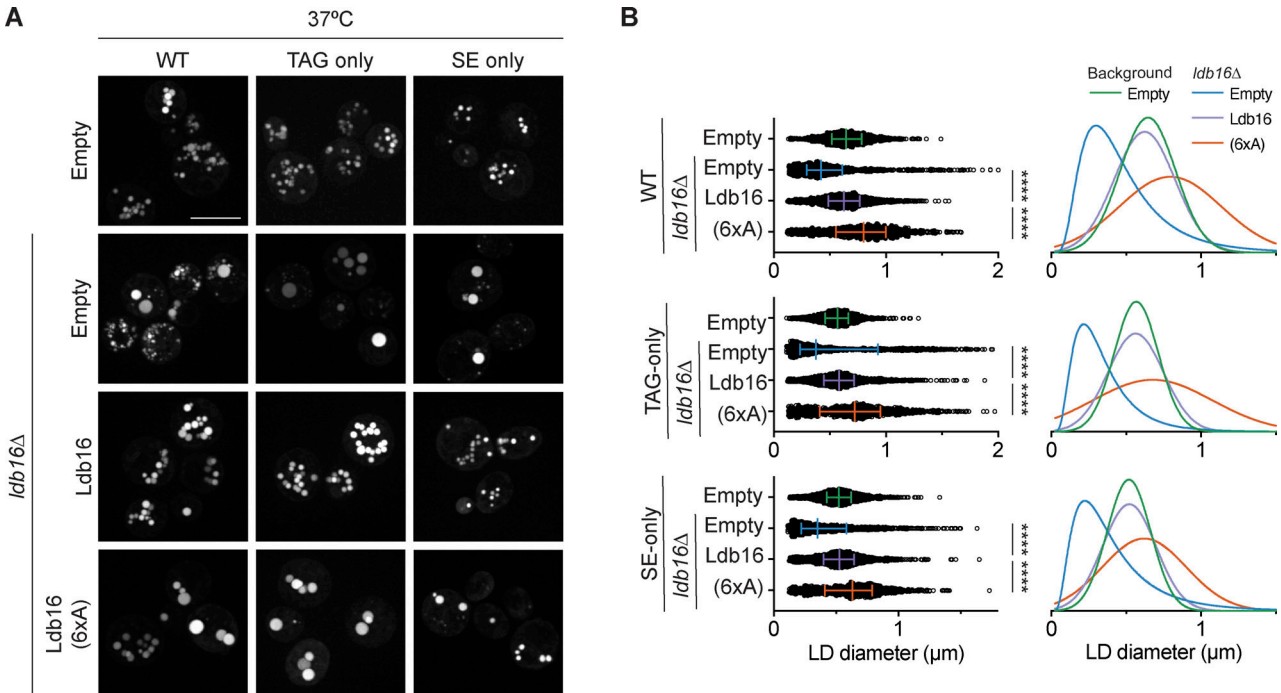

**Figure 7. Hydroxyl-containing residues in yeast Ldb16 are required for correct LD morphology in TAG-only and SE-only cells. (A)** Analysis of LD morphology in cells with the indicated genotype by fluorescence microscopy. Cells were cultured in inositol-free media at 37°C to late stationary phase, and LDs were stained with BODIPY. Scale bars correspond to 5 μm. **(B)** Quantification of LD size in cells with indicated genotype grown as in A. Left: Individual LD diameter measurements. The difference in distribution of LD size was analyzed by non-parametric testing (Kruskal–Wallis test) followed by Dunn's multiple comparison analysis (**, P < 0.01, ****, P < 0.0001). Right: Non-linear regression fitted curves of histograms of LD diameter frequency distribution (bins 0.05 μm). Non-linear regression tested fits of normal (Gaussian) or log-normal curves and the most probable fit (>99.99% certainty) is shown.

yeast nitrogen base (MP Biomedicals), 0.6 g complete supplement mix without his/leu/met/ura (MP Biomedicals), 20 g raffinose (Sigma-Aldrich), and was supplemented with amino acids as required.

**Light microscopy**

For microscopy analysis of LD formation and morphology, cells were cultured as described and LDs were visualized with BODIPY 493/503 (1 μg/ml; Invitrogen).

Super-resolution spinning disc confocal microscopy was performed on an Olympus IX-83 inverted frame confocal microscope, equipped with a Yokogawa CSU-W1 SoRa super-resolution spinning disc module, and a Photometrics Prime BSI camera. Images were acquired using an UplanApo 60× objective (N.A. 1.50). Total magnification was 192×. Fluorophores were excited using a 488 nm (BODIPY, mNeonGreen) or 561 nm (mCherry) solid-state laser (OBIS), and fluorescence emission was selected using a 525/50 nm (BODIPY/mNeonGreen) or

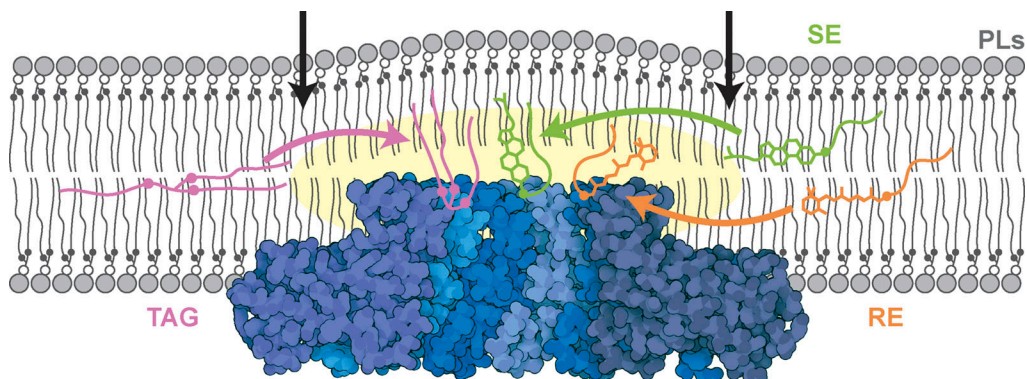

**Figure 8. Model of Seipin enriching NLs via interactions between the hydrophobic helix and NL carboxyl esters.** A model for Seipin-mediated packaging of structurally distinct NLs (TAG, pink; SE, green; RE, orange) into LDs. Seipin luminal domain (PDB accession no. 6DS5, visualized using Illustrate [Goodsell et al., 2019]) protrudes deeply into the ER membrane, positioning its hydroxyl-containing residues far away from PLs carboxyl esters (black circles) and proximal to the bilayer normal, a position that favors its interaction with NLs carboxyl esters (pink, green, and orange circles on TAG, SE, and RE, respectively). This interaction results in NL enrichment and nucleation within the Seipin ring. Arrows indicate the positions of TMs of the outer Seipin protomers.

617/73 nm (mCherry) bandpass filter. Images were processed for super-resolution and deconvoluted (constrained iterative, maximum likelihood) using Olympus cellSens Dimension software (version 3.1.1, build 21264), and the images were exported to TIFF files for further processing.

Widefield epifluorescence microscopy was performed using a Zeiss Axio Observer.Z1 microscope equipped with a Hamamatusu Orca Flash 4.0 digital CMOS camera. Images were acquired using a A Plan APOCHROMAT 100× Objective (N.A. 1.4). BODIPY signal was analyzed using a GFP fluorescence setup consisting of a 485/20 nm bandpass excitation filter (Zeiss), a 410/504/582/669-Di01 quad dichroic mirror, and a 525/30 bandpass emission filter. The microscope was controlled using Slidebook 6.0 software (3i) and acquired images were exported to TIFF files.

Figure preparation and quantification of LD size and number were done using ImageJ (version 1.53c; National Institutes of Health), and no non-linear adjustments were made. For quantification of LD size from super-resolution images, z-max intensity projections were segmented using auto-thresholding, followed by watershedding of the binary image to separate overlapping LDs. Segmented LDs were analyzed using the particle analysis function (size 0.01—infinity), and ellipses were fitted on each particle to determine the diameter of segmented LDs. A circularity selection of 0.8–1.0 was used to exclude overlapping LDs and aberrant-shaped structures.

Differences in LD size distribution were compared by non-parametric testing using Kruskal–Wallis test followed by Dunn's multiple comparison test. In addition, LD size distributions were compared by conversion to histograms (bin width = 0.05 μm) followed by non-linear regression analysis using least squares regression without weighting. Fits of Gaussian (normal) and log-normal distributions were compared using Akaike's Information Criterion to select the model most likely to have generated the data.

### Protein gel electrophoresis and immunoblotting
For protein analysis, cell pellets corresponding to 1 OD unit were harvested by centrifugation and stored at –20°C until further processing. Proteins were extracted using alkaline treatment and heating in sample buffer as described (Kushnirov, 2000). Briefly, cell pellets were resuspended in 0.15 M NaOH and incubated on ice for 10 min. Cells were harvested, resuspended in 1.5× Laemmli sample buffer (Laemmli, 1970), and heated at 65°C with vigorous shaking. Debris was pelleted by short centrifugation, and the samples were loaded on a 4–20% gradient SDS-polyacrylamide gel (Bio-Rad), separated by gel electrophoresis, and blotted on a PVDF membrane. Proteins were analyzed using horseradish peroxidase anti-FLAG (FLAG-HRP; Sigma-Aldrich), anti-Pln1 (Grippa et al., 2015) or anti-Dpm1 primary (Thermo Fisher Scientific), and horseradish peroxidase-conjugated secondary antibodies (Jackson Laboratories).

### Neutral lipid analysis by thin layer chromatography
Cells were cultured to late stationary phase, harvested by centrifugation, washed in cold MQ water, and pellets were stored at –20°C until further processing. Cells were resuspended in 155 mM ammonium formate and lysed by bead bashing (5 × 30 s with 1 min cooling on ice in between cycles). Lipids were

extracted using a modified version of the two-step Bligh&Dyer extraction (Bligh and Dyer, 1959). Briefly, 200 μl of cell lysate was added to 750 μl chloroform/methanol (2:1) and vigorously shaken for 30 min. Then 250 μl chloroform and 250 μl 155 mM ammonium formate were added to induce phase separation, and samples were vigorously shaken for 15 more min. Samples were briefly centrifuged (2 min at 2,500 $g$) and the organic phase was collected. The aqueous phase was re-extracted with 500 μl chloroform for 15 min and the organic phases were pooled. Lipid extracts were dried in a speedvac for 2 h and lipids were dissolved in 25 μl chloroform.

Neutral lipids were separated by thin layer chromatography on silica gel 60 plates using hexane/diethylether/acetic acid (70/30/2, vol/vol/vol) as the eluent (Schneiter and Daum, 2006), ran to ~3/5 of the plate. Plates were dried and re-eluted using hexane/chloroform (9/1, vol/vol) ran to ~4/5 of the plate, to further separate steryl esters, squalene, and hydrocarbons. Lipids were visualized by Cu-staining (10% CuSO$_4$ [wt/vol] in 8% H$_3$PO$_4$) and charring.

### Lipidomics analysis
Cells were cultured to late stationary phase, harvested by centrifugation, and washed twice with MQ water, snap frozen, and stored at –80°C. Cell pellets were resuspended in MQ at 20 OD/ml and lysed by bead bashing 5 × 30 s with 1 min cooling on ice in between cycles. Lysates were snap-frozen and kept at –80°C or on dry ice until further processing.

Lipid extraction and lipidomics analysis by shotgun mass spectrometry was performed by Lypotype GmbH as described (Klose et al., 2012), using a hybrid quadrupole/Orbitrap mass spectrometer equipped with an automated nano-flow electrospray ion source. Lipid identification was done using LipidXplorer (Herzog et al., 2011).

### Coarse-grained molecular dynamics simulations
Coarse-grained systems were built using the human Seipin luminal domain (PDB accession no. 6DS5), with the TM regions modeled as per the yeast Sei1 TMs (PDB accession no. 7OXP) using Swiss-Model (Waterhouse et al., 2018), with the final model available at osf.io/5depa. The proteins were described using MARTINI 2.2 (Marrink et al., 2007; Monticelli et al., 2008). Elastic networks of 500 kJ mol$^{-1}$ nm$^{-2}$ were applied between all protein backbone beads within 1 nm. Distance restraints with flat-bottomed potentials of 1,000 kJ mol$^{-1}$ nm$^{-2}$ were applied using PLUMED 2.4.4 (PLUMED consortium, 2019) between the backbone beads of V248 of one protomer, with L39, L167, and V248 of the next protomer to maintain an appropriate quaternary structure of the TM regions.

Proteins were built into membranes using the insane protocol (Wassenaar et al., 2015) with membrane compositions as listed in Table S3. Equivalent systems without Seipin were also built. SE and TAG parameters are from Vuorela et al. (2010), RE are from Molenaar et al. (2021), and SQL was parameterized for this study (see the section below) and is available at https://osf.io/wnzsf/. These parameters were built based on bond angles and lengths extracted from ~250 ns of atomistic simulation (see the section below).

All systems were solvated with MARTINI waters and Na$^+$ and Cl$^-$ ions to a neutral charge and 0.15 M. Systems were minimized using the steepest descents method, followed by 1 ns equilibration with 5-fs time steps, then by a 100-ns equilibration with 20-fs time steps before 3 × 5 µs production simulations using 20-fs time steps, all in the NPT ensemble with the V-rescale thermostat at 323 K and semi-isotropic Parrinello-Rahman pressure coupling (Bussi et al., 2007; Parrinello, 1981).

Simulations were run in Gromacs 2018 (Abraham et al., 2015) patched with PLUMED 2.4.4 (PLUMED consortium, 2019). Images were made in VMD (Humphrey et al., 1996). Lipid analyses were run using Gromacs analysis tools and the *PyLipID* package (Song et al., 2022). Lipid enrichment by Seipin was quantified as the number of each lipid within 6 nm of the centre-of-geometry of the Seipin complex as a product of total lipids present in the system. Cluster sizes were calculated using the Gromacs tool *gmx clustsize* with a cutoff of 0.6 nm based on the final 3 µs of a 5 µs simulation.

### Paramaterization of SQL

To parameterize the SQL lipid, an atomistic system was built of SQL in a *sn*-1-palmitoyl-*sn*-2-oleoyl-phosphatiylcholine (POPC) membrane with TIP3P waters and K$^+$ and Cl$^-$ at a concentration of 150 mM using CHARMM-GUI (Jo et al., 2007; Lee et al., 2016) and using the CHARMM-36m force field (Best et al., 2012; Huang et al., 2017). Simulations were run with 2-fs time steps over 250 ns in the NPT ensemble with the V-rescale thermostat at 298 K and semi-isotropic Parrinello-Rahman pressure coupling (Bussi et al., 2007; Parrinello, 1981). Each isoprene unit was virtually mapped as a MARTINI bead, and the distances and angles between adjacent beads were extracted from the simulation. The mean values for these were used to generate the MARTINI SQL parameters, see https://osf.io/wnzsf/, with force constants set to achieve similar distributions between the CG and AT parameters. Bonded force constants were checked against experimental *logP* values using decoupling of the lipid in a box containing 878 MARTINI waters or 415 octanol and 34 waters. Lipids Lennard-Jones parameters were decoupled over 21 evenly-spaced λ windows, for 50 ns per window. Simulations were run with 20-fs time steps in the NPT ensemble with the V-rescale thermostat at 323 K and an isotropic Parrinello-Rahman pressure coupling (Bussi et al., 2007; Parrinello, 1981).

### Online supplemental material

Fig. S1 provides additional information on TAG-only and SE-only cells and describes the pipeline used to analyze LD size. Fig. S2 shows the effect of temperature on LD composition in TAG-only and SE-only cells and the impact of mutations in the Seipin complex on the morphology of RE-only LDs. Fig. S3 shows the interactions between human Seipin and lipids using MD simulations. Fig. S4 shows the effects of human Seipin in the formation of TAG-only and SE-only LDs. Fig. S5 shows the effects of yeast complex subunit Ldb16 in the formation of TAG-only and SE-only LDs. Table S1 contains the yeast strains used in this study. Table S2 contains the plasmids used in this study. Table S3 contains the molecular dynamic simulations performed in this study.

## Acknowledgments

We thank Y. Klug for valuable discussions and critical reading of the manuscript, R. Ernst for critical reading of the manuscript, T. Wassenaar for sharing the retinyl ester MD parameters, M. Hermansson for expertise on lipid extraction procedures, A. Wainman and E. Johnson for technical assistance with microscopy, and C. Peselj for helpful suggestions on image analysis using Fiji/ImageJ. Super-resolution microscopy imaging was conducted at the Dunn School Bioimaging facility.

P. Carvalho's lab is supported by a Biotechnology and Biological Sciences Research Council grant (BB/R018375/1) and an investigator award from Wellcome (202642/Z/16/Z). R.A. Corey and P.J. Stansfeld are supported by Wellcome (208361/Z/17/Z). P.J. Stansfeld's lab is supported by awards from the Biotechnology and Biological Sciences Research Council (BB/P01948X/1, BB/R002517/1, and BB/S003339/1) and Medical Research Council (MR/S009213/1). P.J. Stansfeld acknowledges the University of Warwick Scientific Computing Research Technology Platform for computational access.

The authors declare no competing financial interests.

Author contributions: M.F. Renne and P. Carvalho conceptualized the study. M.F. Renne performed most of the experiments, with help of J.V. Ferreira. R.A. Corey performed molecular dynamics simulations guided by P.J. Stansfeld. M.F. Renne and R.A. Corey visualized data. P. Carvalho supervised the project. M.F. Renne and P. Carvalho wrote the draft manuscript. All authors commented on drafts of the manuscript.

Submitted: 13 December 2021

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

# Supplemental material

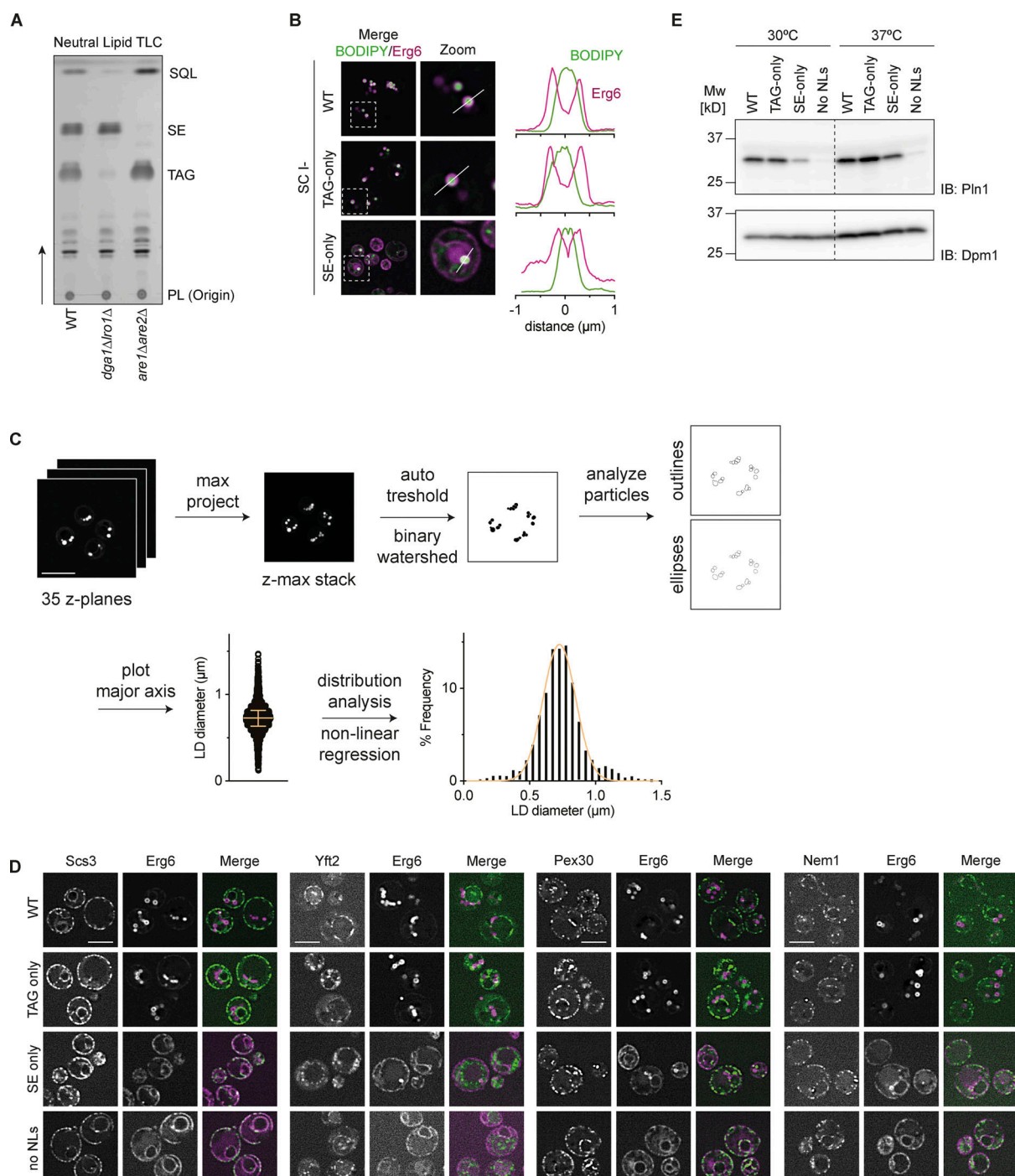

Figure S1. **Characterization of lipid droplets in WT, TAG-only, SE-only cells.** Related to Fig. 1. **(A)** Neutral lipid analysis by thin layer chromatography for WT, *dga1Δlro1Δ* (SE-only) and *are1Δare2Δ* (TAG-only) cells. Spots corresponding to phospholipids (PL), triacylglycerol (TAG), sterol esters (SE) and squalene (SQL) were identified based on $R_f$-values of previously run standards. **(B)** Left: Fluorescence micrographs of LDs in WT, *are1Δare2Δ* (TAG-only), *dga1Δlro1Δ* (SE-only) cells cultured to late stationary phase in SC-ino media as in Fig. 1 A. Merge of BODIPY (NL stain) and Erg6 (LD marker protein) is shown. Boxes (5 × 5 µm) indicate location of zoom blow-up images (middle). Line in the zoom images indicates location of line intensity plots (right). **(C)** Schematic overview of LD size quantification. LDs were stained with BODIPY and imaged by super resolution fluorescence microscopy (35 z-planes, total depth 8.4 µm). Z-planes were projected for maximum intensity, and LDs were segmented by thresholding yielding binary images. Overlapping LDs were separated using the binary watershed function and analyzed using the particle analysis function with ellipses being fitted on all particles (circularity cut off 0.8–1.0). The major axis of the fitted ellipses was used to determine LD diameter. The yielded LD diameter distribution was converted to histograms using distribution analysis, and non-linear regression analysis compared the fit of gaussian (normal) or Log-normal distribution. **(D)** Localization of Scs3, Yft2, Pex30, and Nem1 in WT, TAG-only, SE-only and no LDs cells. Proteins of interest were expressed from their endogenous loci as C-terminal fusions to mNeonGreen and Erg6-mCherry was used as LD marker. Scale bars correspond to 5 µm. **(E)** Expression levels of Pln1 in indicated mutants cultured to late stationary phase at 30 or 37°C. Whole cell lysates were separated by SDS-PAGE and analyzed by immunoblot with anti-Pln1 antibody. Dpm1 was used as a loading control and detected with an anti-Dpm1 antibody. Source data are available for this figure: SourceData FS1.

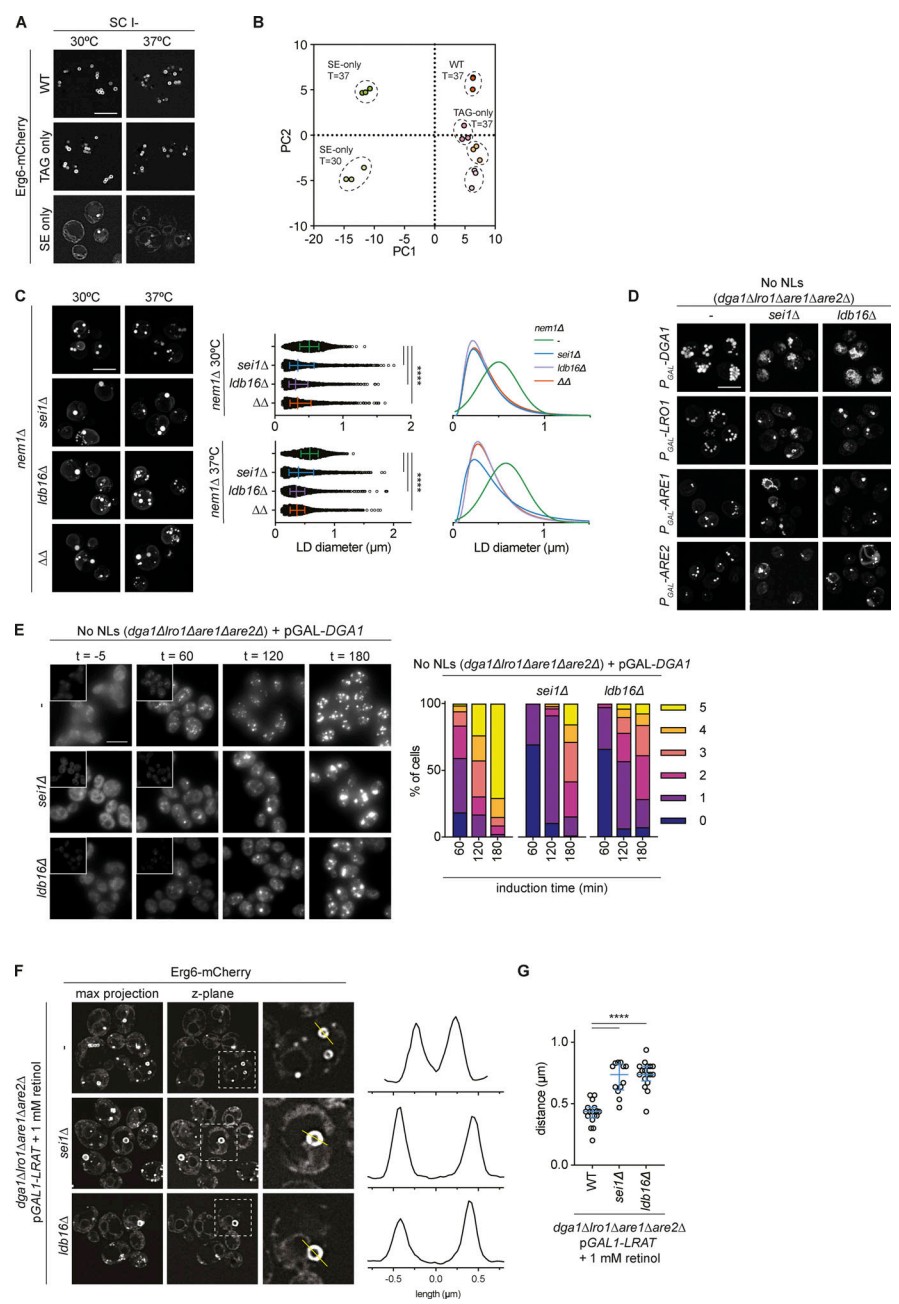

Figure S2. **Sei1/Ldb16 Seipin complex is required for normal morphology and biogenesis of TAG-only and SE-only LDs.** Related to Figs. 2 and 3. **(A)** Fluorescence micrographs of LDs in WT, TAG-only and SE-only cells cultured in inositol free media to late stationary phase at indicated temperature. Erg6 serves as LD marker protein, scale bars correspond to 5 μm. **(B)** Principal Component Analysis (PCA) of lipid molecular species composition (in mol%) of indicated mutants. Proportion of variance is 58.25% (PC1) and 13.17% (PC2). **(C)** Analysis of LD morphology in *nem1Δ* background strains with indicated genotypes. Cells were cultured in inositol free media at 30 or 37°C to late stationary phase, and LDs were stained with BODIPY. Scale bars correspond to 5 μm. **(D)** Analysis of LD morphology in no NL cells upon plasmid borne overexpression of the indicated NL synthesizing enzyme. Micrographs were taken after 24 h addition of galactose to induce the expression of the NL synthesizing enzymes. LDs were visualized using the NL dye BODIPY 493/503. Scale bars correspond to 5 μm. **(E)** Analysis of LD biogenesis upon induction of TAG synthesis by expression of the DAG acyltransferase Dga1. Plasmid-borne expression of Dga1 was induced by addition of galactose (final concentration 2%) to cells with the indicated genotype grown in raffinose-containing medium. LDs were stained with BODIPY at indicated timepoints (in minutes after galactose induction). Scale bars correspond to 5 μm. The number of LDs per cell was quantified for a minimum of 50 cells per timepoint. **(F)** Analysis of LD morphology in cells with indicated genotypes, expressing plasmid-borne human LRAT from the *GAL1* promotor. Left: Fluorescence micrographs of Erg6-mCherry (z-max projections and single z-planes) are shown. Dashed boxes indicate location of blow-up images (8 × 8 μm), location of intensity profile is indicated by yellow lines. Right: Line intensity plot of Erg6-mCherry signal. Distance between peak maxima is indicative of LD size. LRAT expression was induced by the addition of galactose (final concentration 2%) and retinol was supplemented at 1 mM (in 1% Tergitol NP-40). LDs were visualized using Erg6-mCherry as a LD marker protein after 8 h of LRAT induction and retinol supplementation. Clustered and supersized LDs typical of *sei1Δ* and *ldb16Δ* mutants are indicated by red and yellow arrowheads, respectively. **(G)** Distance distribution of LD size as determined by Erg6-mCherry line intensity analysis (as in F). Circles show individual measurements, line indicates median (± 95% confidence interval). Difference in distributions were analyzed by non-parametric testing (Kruskal–Wallis test; P < 0.0001), followed by Dunn's multiple comparison testing (****, P < 0.0001).

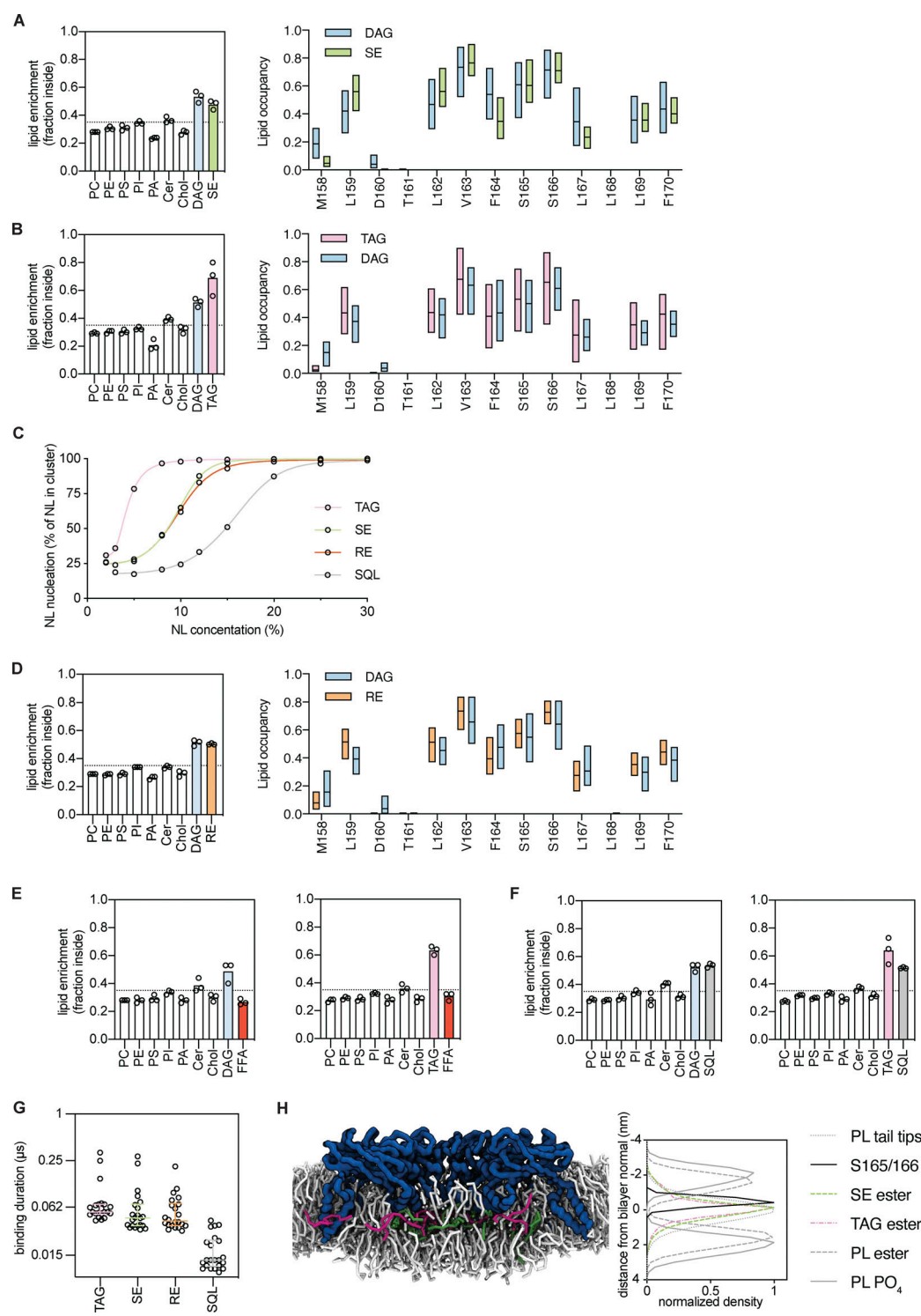

Figure S3. **Seipin enriches TAG and SE via interactions with its hydrophobic helix.** Related to Figs. 4 and 5. **(A and B)** Analysis of lipid enrichment by human Seipin (left), shown as fraction of total lipids inside the Seipin ring, and analysis of human Seipin HH residues occupancy (right) by SE (A) or TAG (B) as a single NL present. For lipid enrichment (left), bars represent the mean of three independent simulations (circles). For HH occupancy, (right) bars min to max values with a line at the median. **(C)** Analysis of NL clustering dependency on NL concentration. Individual datapoints are shown, a five-point sigmoidal curve is fitted to guide the eye. **(D)** As (A and B) but analysis of RE enrichment by human Seipin (left), shown as fraction of total lipids inside the Seipin ring, and analysis of human Seipin HH residues occupancy (right). **(E and F)** Analysis of lipid enrichment by human Seipin in simulations containing FFA (E) or SQL (F) in the absence (left) or presence (right) of TAG. Dashed line indicates the fraction of the membrane surface occupied by the Seipin ring. Bars represent the mean of three independent simulations (circles). **(G)** Analysis of the NL-binding time to residues S165/166 of human Seipin. Binding time was measured using *PyLipid* package using a double cut off of 0.55 and 1.0 µm. Individual measurements are shown, bar and whiskers indicate median ± interquartile values. **(H)** Snapshot of TAG/SE enrichment by Seipin (left) and density analysis of selected groups along the bilayer perpendicular (z-axis; right). Density was normalized with maximum density set at 1. Bilayer normal (z = 0 nm) was defined as the z-coordinates with the maximum density for the PL acyl chain tips.

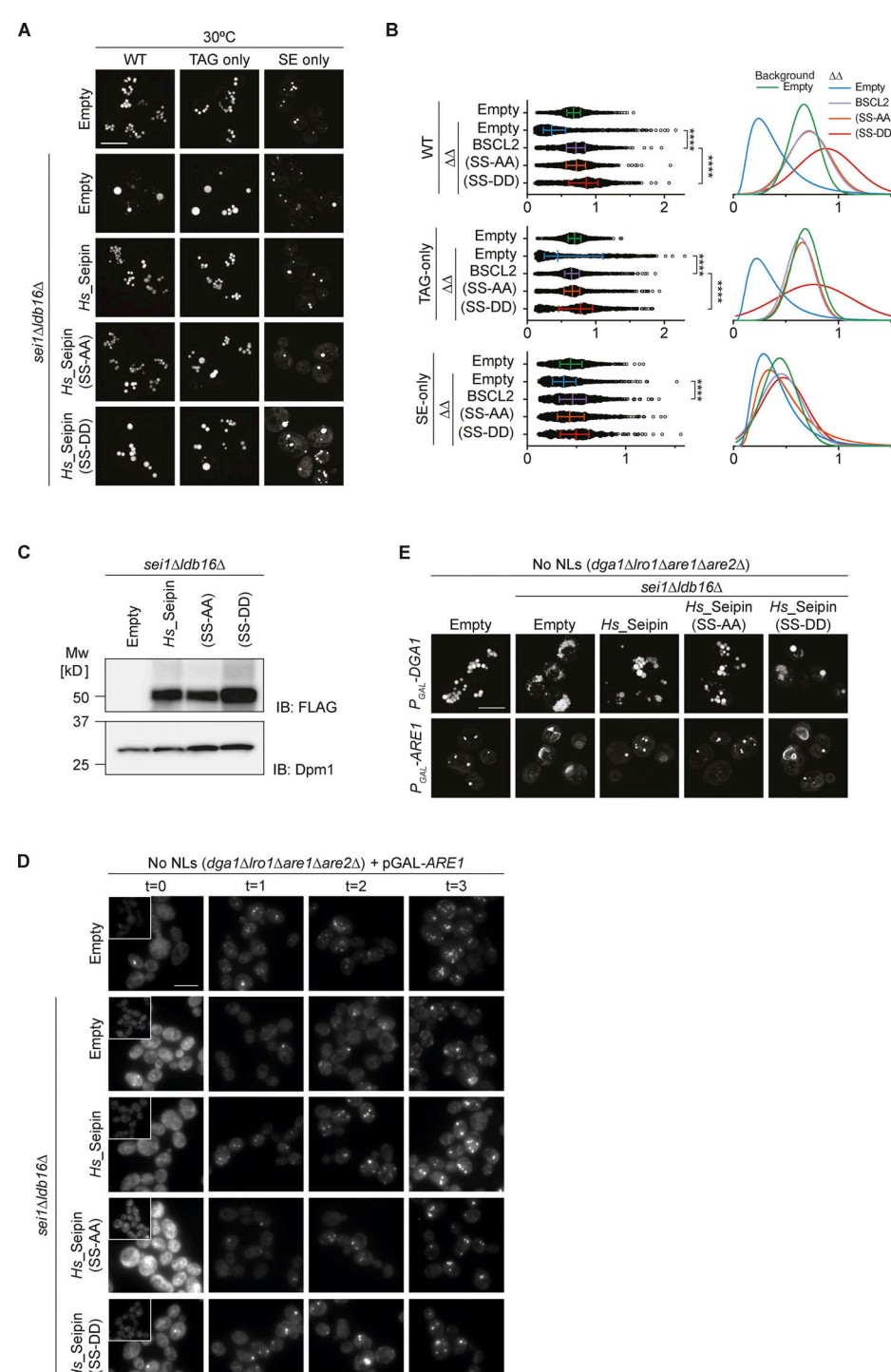

Figure S4.    **Hydroxyl-containing residues in human Seipin are required for correct LD morphology in TAG-only and SE-only cells.** Related to Fig. 6.
**(A)** Analysis of LD morphology in cells with indicated genotype by super resolution fluorescence microscopy. Cells were cultured in inositol free media at 30°C to late stationary phase, and LDs were stained with BODIPY. Scale bars correspond to 5 µm. **(B)** Quantification of LD size in cells shown in A. A minimum of 100 LDs per genotype and two independent biological replicates were quantified. Bars and whiskers indicate mean with interquartile range. **(C)** Levels of FLAG-tagged WT human Seipin and its derivatives with mutations in S165/166, SS-AA and SS-DD, expressed in *sei1Δldb16Δ* cells. Whole cell lysates were separated by SDS-PAGE and analyzed by immunoblot with anti-FLAG antibody. Dpm1 was used as a loding control and detected with an anti-Dpm1 antibody. **(D)** Analysis of LD biogenesis upon induction of SE synthesis by expression of the SE acyltransferase Are1. Plasmid-borne expression of Are1 was induced by addition of galactose (final concentration 2%) to cells with the indicated genotype grown in raffinose-containing medium. LDs were stained with BODIPY at indicated timepoints (in hours after galactose induction). Scale bars correspond to 5 µm. **(E)** Analysis of LD morphology in cells with indicated genotype upon plasmid borne overexpression of the NL enzymes Dga1 or Are1 which synthesize TAG or SE, respectively. Micrographs were take after 24 h addition of galactose to induce the expression of the NL synthesizing enzymes. LDs were visualized using the NL dye BODIPY 493/503. Scale bars correspond to 5 µm. Source data are available for this figure: SourceData FS4.

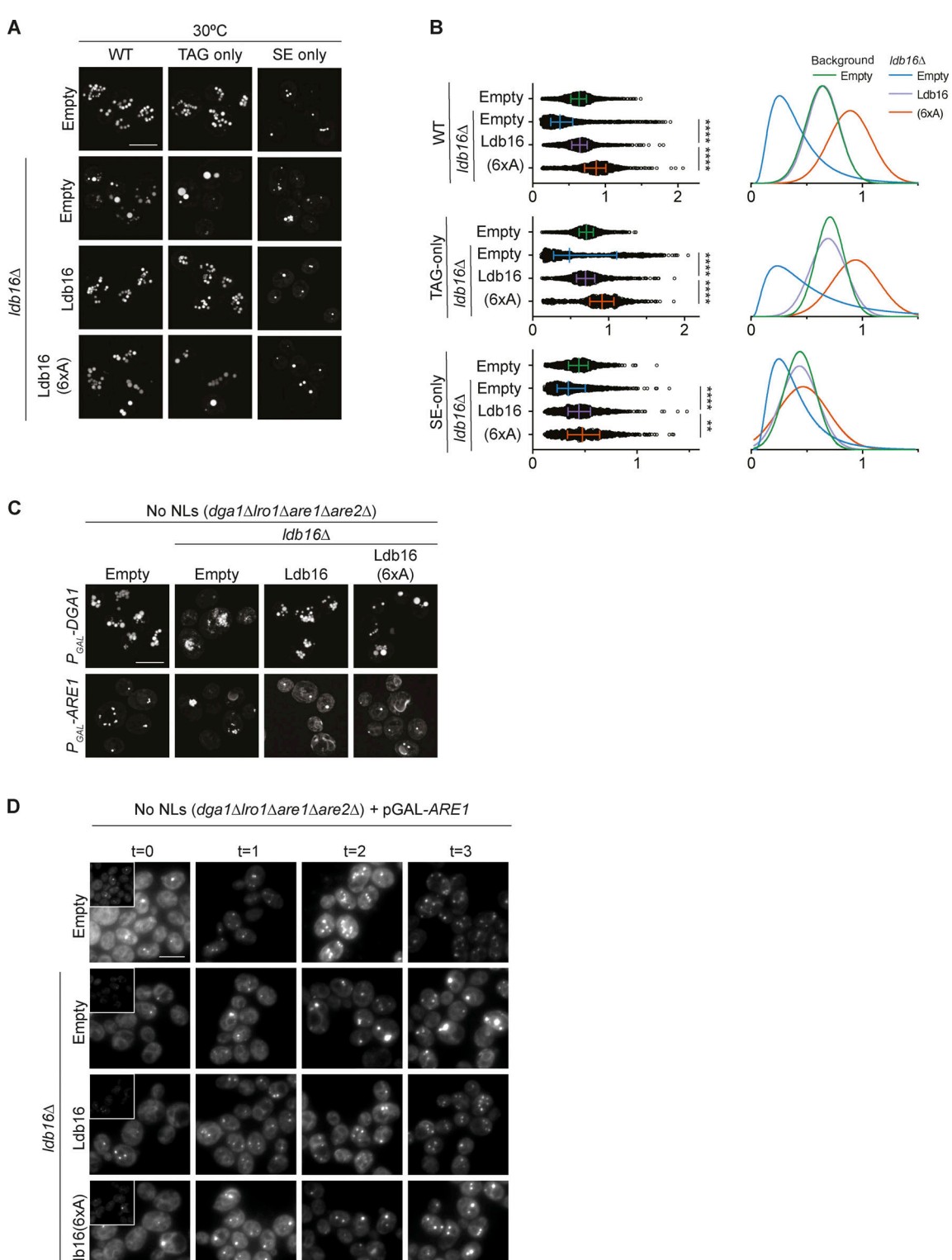

Figure S5. **Hydroxyl-containing residues in yeast Ldb16 are required for correct LD morphology in TAG-only and SE-only cells.** Related to Fig. 7. **(A)** Analysis of LD morphology in cells with the indicated genotype by fluorescence microscopy. Cells were cultured in inositol free media at 30°C to late stationary phase, and LDs were stained with BODIPY. Scale bars correspond to 5 μm. **(B)** Quantification of LD size in cells shown in A. A minimum of 100 LDs per genotype and two independent biological replicates were quantified. Bars and whiskers indicate mean with interquartile range. **(C)** Analysis of LD morphology in cells with indicated genotype upon plasmid borne overexpression of the NL enzymes Dga1 or Are1 which synthesize TAG or SE, respectively. Micrographs were take after 24 h addition of galactose to induce the expression of the NL synthesizing enzymes. LDs were visualized using the NL dye BODIPY 493/503. Scale bars correspond to 5 μm. **(D)** Analysis of LD biogenesis upon induction of SE synthesis by expression of the SE acyltransferase Are1. Plasmid-borne expression of Are1 was induced by addition of galactose (final concentration 2%) to cells with the indicated genotype grown in raffinose-containing medium. LDs were stained with BODIPY at indicated timepoints (in hours after galactose induction). Scale bars correspond to 5 μm.

**Provided online are Table S1, Table S2, and Table S3. Table S1 list of yeast strains used in this study. Table S2 list of plasmids used in this study. Table S3 shows coarse grained MD simulations run in this study.**

