## [Peer Review File · The Journal of Cell Biology]

Seipin concentrates distinct neutral lipids via interactions with their acyl chain carboxyl esters

Mike Renne, Robin Corey, Joana Veríssimo Ferreira, Phillip Stansfeld, and Pedro Carvalho

Corresponding Author(s): Pedro Carvalho, University of Oxford

Review Timeline:

Submission Date:	2021-12-13
Editorial Decision:	2022-01-18
Revision Received:	2022-05-28
Editorial Decision:	2022-07-05
Revision Received:	2022-07-11

Monitoring Editor: Jodi Nunnari

Scientific Editor: Andrea Marat

Transaction Report:

DOI: <https://doi.org/10.1083/jcb.202112068>

January 18, 2022

Re: JCB manuscript #202112068

Dr. Pedro Carvalho
University of Oxford
Sir William Dunn School of Pathology
South Parks road
South Parks Road
Oxford, UK OX1 3RE
United Kingdom

Dear Dr. Carvalho,

Thank you for submitting your manuscript entitled "Seipin concentrates distinct neutral lipids via interactions with their acyl chain carboxyl esters". The manuscript was assessed by expert reviewers, whose comments are appended to this letter. We invite you to submit a revision if you can address the reviewers' key concerns, as outlined here.

As you will see, the reviewers appreciate the potential impact of your findings. However, they have indicated that a robust quantification analysis of your LD images, which includes appropriate statistical measures, is essential to support your conclusions. In addition, additional analyses, such as the lipid measurements, suggested by reviewer 3 are also required to support your conclusions. Please also address the points and questions raised by reviewer 2.

Although we understand your study was submitted as a Report given its descriptive nature, the main advance is a fundamental new insight into the process of LD formation. We recognize that a significant amount of work will be required to address the reviewers' concerns for resubmission to JCB. Thus, editorially we find it suitable as an Article, which will provide you additional space to address the reviewers' concerns in full.

GENERAL GUIDELINES:

Text limits: Character count for an Report is < 40,000, not including spaces. Count includes title page, abstract, introduction, results, discussion, acknowledgments, and figure legends. Count does not include materials and methods, references, tables, or supplemental legends.

Figures: Reports may have up to 10 main text figures. Figures must be prepared according to the policies outlined in our Instructions to Authors, under Data Presentation, <https://jcb.rupress.org/site/misc/ifora.xhtml>. All figures in accepted manuscripts will be screened prior to publication.

Supplemental information: There are strict limits on the allowable amount of supplemental data. Reports may have up to 5 supplemental figures. Up to 10 supplemental videos or flash animations are allowed. A summary of all supplemental material should appear at the end of the Materials and methods section.

Please note that JCB now requires authors to submit Source Data used to generate figures containing gels and Western blots with all revised manuscripts. This Source Data consists of fully uncropped and unprocessed images for each gel/blot displayed in the main and supplemental figures. Since your paper includes cropped gel and/or blot images, please be sure to provide one Source Data file for each figure that contains gels and/or blots along with your revised manuscript files. File names for Source Data figures should be alphanumeric without any spaces or special characters (i.e., SourceDataF#, where F# refers to the associated main figure number or SourceDataFS# for those associated with Supplementary figures). The lanes of the gels/blots should be labeled as they are in the associated figure, the place where cropping was applied should be marked (with a box), and molecular weight/size standards should be labeled wherever possible.

As you may know, the typical timeframe for revisions is three to four months. However, we at JCB realize that the implementation of social distancing and shelter in place measures that limit spread of COVID-19 also pose challenges to scientific researchers. Lab closures especially are preventing scientists from conducting experiments to further their research. Therefore, JCB has waived the revision time limit. We recommend that you reach out to the editors once your lab has reopened to decide on an appropriate time frame for resubmission. Please note that papers are generally considered through only one revision cycle, so any revised manuscript will likely be either accepted or rejected.

Thank you for this interesting contribution to Journal of Cell Biology. You can contact us at the journal office with any questions, cellbio@rockefeller.edu or call (212) 327-8588.

Sincerely,

Jodi Nunnari, Ph.D.
Editor-in-Chief

Andrea L. Marat, Ph.D.
Senior Scientific Editor

Journal of Cell Biology

Reviewer #1 (Comments to the Authors (Required)):

The study examines how human Seipin and yeast Sei1 and its cofactor Ldb16 contribute to the formation of LDs containing TAG or SE. Yeast that form TAG-only or SE-only LDs appear to show LD morphological defects when Sei1 or Ldb16 are deleted. They also show defects in yeast forming retinoyl ester LDs when sei1 is missing. Molecular modeling data suggests that the same regions of Seipin that bind to TAG can also help to corral SE for LD formation. Whereas human Seipin can rescue sei1 knockout yeast, mutation of the previously characterized TAG binding residues perturbs LD morphology in TAG-only and SE-only strains, suggesting Seipin is required for proper corralling of TAG and SE in LD formation. In yeast, Ldb16 has been proposed as engaging the neutral lipids for LD formation. It is suggested that a mutant Ldb16 causes LD defects in TAG-only and SE-only LD formation, suggesting it is needed for LD formation of both LD compositions.

The study addresses an important question in the field. However, a major concern is how LD morphology and diameter are being analyzed. It is unclear what statistical approaches are implemented to compare LD diameter among different conditions. Without rigorous statistical analysis, it appears that some of the conclusions are not supported by the data. This is a problem throughout the work. Addressing this is crucial before any major conclusions can be made as to some mutant rescues.

concerns

a. In Figure 4B/D for example, there is no indication for statistical analysis with Seipin mutants. It is stated that Seipin SS-AA and SS-DD mutants fail to rescue LD diameter whereas Seipin WT does rescue. The SS-AA and SS-DD look quantitatively very different from one another, so some additional analysis here is required.

b. Similarly in Figure 2 B/D, where are no statistics presented, but at 30C it appears that there is no difference in LD diameter in sei1 or ldb16 knockout yeast for SE-only droplets. This should be stated in the text. The 37C experiment does appear to show a clear difference in the representative images, but the LD diameter quantification appears to not show any difference. This should be analyzed with statistics and reconciled.

c. A general conclusion is that sei1 and ldb16 are required for the formation of TAG-only or SE-only LDs. However, in several images presented here, it appears that SE-only LDs do not contain the same complement of proteins as TAG-only LDs. In F1, Erg6 does not decorate SE-only LDs like the TAG-only LDs. An alternative model for the study is that SE-only LDs contain a different complement of surface proteins, which affects their stability and morphology. Some discussion is necessary.

Reviewer #2 (Comments to the Authors (Required)):

The authors have used cell experiments (yeast) and coarse-grained (CG) MARTINI model simulations to study how seipin concentrates distinct neutral lipids (TAG, SE, RE, and SQL). While the manuscript contains valuable findings and may advance the understanding of lipid droplet formation (especially, with various neutral lipids), the authors should first address the major and minor concerns written below.

MAJOR CONCERNS

1. The authors claimed that formation of hydrogen bonds between carboxyl esters of neutral lipids and S165/S166 is required for lipid droplet formation. For instance,

- "The TAG concentrating activity of human Seipin depends on the HH, requiring two serine residues (S165/166) in this helix" on Page 3, line 31.

- "This activity involved the formation of a hydrogen bonds (H-bonds) between conserved hydroxyl-containing residues in the central HH and carboxyl esters present in TAG and SE." on Page 4, line 54.

However, this is not supported by the provided data, and I would soften those conclusions and sentences (perhaps replace S165/S166 with hydrophobic helix?). For instance,

- Distribution of LD diameters in SS-AA (S165 and S166 were mutated to alanine)-containing cells and pSeipin-containing cells are very close in Fig. 4B. Accordingly, I would suggest updating the following sentence on Page 9, line 201. "Importantly, mutation of Seipin S165/166 either to alanine (SS-AA) or aspartate (SS-DD) failed to restore normal LD morphology in sei1Δldb16Δ cells"

- Squalene that does not have any hydrogen donors enriches in the seipin ring (Fig. S2E).

- Fatty acids that do have hydrogen donors do not enrich in the seipin ring (Fig. S2D).

Also, I would suggest that the authors be more cautious about the usage of hydrogen bonds when analyzing the MARTINI simulations. Strictly speaking, there are no explicit "hydrogen" atoms in MARTINI model coarse-grained simulations. Also, the physical limitations of the MARTINI CG model should be mentioned and cited (see Jarin et al., J. Chem. Theory Comp. 17, 1170-1180 (2021)). This model should not be used blindly.

2. While I am not an imaging expert, I find the following concerns about Fig. 1A and S1G.

- Figure 1A: Generally speaking, BODIPY 493/503 is known for intercalating with TAG and BODIPY-CE with SE. How do the authors know BODIPY 493/503 is a good SE marker?

- Figure 1A-YPD-SE-only-BODIPY: Why is there a ring-like intensity in the leftmost lipid droplet? You might need to zoom in this figure to see this. Is there a phase separation as shown in Julia Mahamid et al., PNAS, 2019, 116 (34)? See Fig S3A of this paper.

- Erg6 seems to target TAG-LD but does not target SE-LD in Fig. 1A. Given this concern, how do the authors argue that Erg6 is a good LD marker for retinyl esters LD in Fig. S1G? Also, considering this, Erg6 should be referred to as the TAG-LD marker protein rather than simply the LD marker protein.

3. The authors made new molecules (SE, RE, and SQL) in the MARTINI simulations. Have the authors created "new" atomic types that are not included in the original release of MARTINI? If yes, please indicate all the details. Otherwise, the atomic types used should be clearly indicated in Methods and Fig 3.

Page 14, line 371: "These parameters were built based on bond angles and lengths extracted from ca. 250 ns of atomistic simulation of SQL in a sn-1-palmitoyl-sn-2-oleoyl-phosphatidylcholine (POPC) membrane with TIP3P waters and K⁺ and Cl⁻ at a concentration of 150 mM." Were bond and angle parameters parameterized based on all-atom simulations? If so, how? (e.g., Boltzmann Inversion) Also, the simulation details and parameters of the all-atom simulations should be described.

4. The authors calculated the following quantities in their MARTINI simulations: lipid enrichment, lipid occupancy, and binding duration. However, how they characterized those quantities are not described.

- Lipid enrichment (fraction inside): Was lipid composition taken into account when calculating lipid enrichment? For instance, the bilayers are enriched with PC rather than TAG. Therefore, PC concentration is higher than TAG, likely everywhere. Please describe how this effect was normalized. Also, the definition of whether the molecules were included in the seipin ring should be

also described.

- Lipid occupancy: I assume that the authors used a distance-based coordination number or contact number to calculate lipid occupancy. What cutoff distances did the authors use? Please describe this clearly.

- Binding duration: Again, I assume the authors used a distance-based analysis. What cutoff distances did they use?

5. Please indicate atomic types of TAG, RE, and SQL. Regarding atomic types, I have the following concerns:

- Do the following atoms have the same atomic types: C3A, C3B, C3C, and C3D of TAG and C3, C4, C5, C6 of SE? Are C3B and C3C of TAG the same atomic types? If so, the MARTINI force fields that the authors used seem not to consider the double bonds in the acyl chains. Therefore, TAG represents tripalmitin (solid at room temperature), not triolein (the most abundant fat).

- Although there is no direct mapping scheme between all-atom and MARTINI CG force fields, the authors should clearly describe what atomic types were used for the glycerol moieties. For instance, what is GLY of TAG in Fig. 3E? Please note that such a grouping (GLY) does not exist in phospholipids in MARTINI force fields.

6. As SE, RE, and SQL are new molecules (and TAG also, as it was not constructed in a standard way, e.g., inclusion of GLY that is not included in phospholipids and lack of double bonds in acyl chains), the authors should characterize the physical properties of those molecules and compare those with experimental data, e.g., phase at room temperature, density, interfacial tension against water, lattice parameters if solid at room temperature.

7. Coarse-grained simulations have advantages of exploring larger length scales and longer time scales. However, the MARTINI simulations here only showed "recruitment" of neutral lipids but no "nucleation" of neutral lipids. Do neutral lipids eventually nucleate? Given the system sizes shown here, I am certain that the authors can extend the simulations. Please note that all the other computational simulations about seipin showed "nucleation".

8. As a control experiment, simulations of pure lipid systems (no seipin) should be shown as well as to whether neutral lipids nucleate at critical concentrations.

9. Figure S2: There is no system that has both DAG and TAG according to Table S3. Either Figure S2 or Table S3 should be updated.

MINOR CONCERNS

1. Page 4, line 41: "In contrast, foam cells, lipid-laden macrophages found in atherosclerotic lesions, contain LDs that mainly consist of SE". Add references.

2. Figure 1A and Page 5, line 65: Do "quadrouple mutant (QM)" and "no NLs" represent the same cells (*dga1Δlro1Δare1Δare2Δ*)? For consistency, I would use either of them.

3. Figure 2E right: I would add the x label in the plot although it is explained in the legend.

4. Page 7 line 140: Acronym RE was already mentioned in the Introduction.

5. Page 8, line 189: Different insertion depths of PL and TAG carboxyl esters were already shown in bilayers [Campomanes et al., *Biophysical Reports*, 2021, 1(2); Kim and Voth, *JPCB*, 2021, 125(25)]. Cite those papers.

6. Figure S2: For consistency, use either SE or CE.

7. Page 11, line 258: "Both NLs and PLs contain carboxyl ester groups that could in principle accept H-bond from Seipin hydroxyl residues. However, the projection of Seipin HH deep into the membrane bilayer, positions S165/166 close to the NL carboxyl esters at the center of the bilayer and distant from the PL carboxyl esters, which are at the membrane-water interface". The same perspective was discussed in Kim et al., *bioRxiv* 10.1101/2021.12.05.471300. Perhaps, cite this paper?

8. Page 11, line 267: Acronym CCT is not introduced before. Spell it out.

9. Page 14, line 363: What is a flat-bottomed distance restraint? Is it another elastic network?

10. Page 14, line 368: In the initial structures, how many phospholipids were included inside the hydrophobic ring of seipin in the luminal leaflet?

11. Please include page numbers in the manuscript.

Reviewer #3 (Comments to the Authors (Required)):

The submitted manuscript by Renne et al., describes role of yeast seipin Sei1 and its partner protein Ldb16 in the formation of cytoplasmic lipid droplets (LDs). Although the role of these proteins in assembly of triacylglycerol (TAG) rich LDs is known, whether these proteins have important functions in the biogenesis of steryl esters (SE) rich LDs is not completely known. The authors take advantage of yeast genetic mutants that can have exclusively either TAG rich or SE-rich LDs. Previous studies have demonstrated that Seipin is a key protein that define sites of LD formation in the ER. It mainly takes descriptive approaches to try to define if Seipin would have similar role in the biogenesis of SE-containing LDs. The strength of the paper is that some of the findings do advance our knowledge of LD biogenesis. However, most of the data is descriptive and appears preliminary as conclusions are not fully supported by the data (please see specific comments). In my view several things should be addressed:

Specific comments:

1) Figure 1A, why SE-only cells are making less LDs. The authors need to explain this. Is TAG required for nucleation of SE-only LDs? In the same figure see panel SE-only, why Erg6 is not in localizing to few BODIPY puncta that represent SE-LDs. It's not clear the way it is presented.

2) In Fig 1A, panel no NLs, I see a lot of fluorescence of BODIPY in the ER that colocalizes with Erg6. Does this mean the cells have NLs that is dispersed in the ER. Apparently, the cells should have no NLs at all. It needs to be controlled with lipid analysis in this mutant in both culture conditions. As a matter of control see image SE-only, I don't see fluorescence of BODIPY in the ER in cells that do not have any LDs.

3) Fig 1C and D: Since sei1 and ldb16 are part of a complex, it would be advantageous to determine what % of Sei1 and Ldb16 colocalize in cells containing only TAG-LDs, SE-LDs or no-NLs. Fig 1E, western blot to determine if Plin levels are downregulated in cells containing no NLs and compare to other conditions.

4) The authors claim that since SE-only cells have very few LDs, hence it complicated their analysis of LD morphology. However, surprisingly, at 37 they see more SE containing LDs while TAG-containing LDs did not show any change whether grown at 30 or 37. I don't understand this at all. This is an important point. Many experiments are based on this observation. So first the authors should perform lipid analysis to quantitatively determine if SE levels are in-fact higher when cells are shifted to 37, which they claim that the cells are making high SE at 37. The authors should also reason why SE would be higher at 37 and not TAG. Are the two SE-acyltransferases expressed at higher levels at 37.

5) I have several issues with Figure 2, the way authors have interpreted the images and how they have presented. It is well known that cells lacking either Sei1 or Ldb16 have supersized LDs. In Fig 2A-D, I don't see accumulation of supersized LDs in cells containing only SE-containing LDs, though you can clearly see in TAG-containing LDs. Why is it so. Even in the quantification for LD diameter, there is hardly any difference for Wt, sei1, and ldb16 mutants in SE-only LDs. This is central part of the paper. This need to be controlled properly.

6) It will be interesting to add if both sei1 and ldb16 are missing, what would happen to SE-only and TAG-only LDs at 30 and 37. It is important to know if the effect is synergistic. More data are needed to define how it occurs.

7) In S1D, I see clear pattern of the role of Sei1/Ldb16 in LD defects in TAG-containing LDs when cells are stimulated to make de novo LDs. However, comparing the same induction for SE-only LDs, the role of does not seem that apparent. Cells have very few LDs, I don't see clustered or supersized LDs, a classical phenotype of seipin mutants. The majority of cells seem to form at least 1 LD upon 180 min of induction. So it's difficult to attribute a clear role of Sei1 in the biogenesis of SE-only LDs. The authors should perform lipid analysis to see if there is any difference. Similarly, the authors should perform induction with ARE2 to see if it would have dramatic effect of LD morphology defects.

8) Quantification for LD size in S1F is required. Here the authors need to perform lipid analysis to see if at 37 when cells are lacking nem1 together with sei1/Ldb16 are now making more TAG. Earlier they mentioned that at 37 only SE-containing LDs are more and TAG-LDs have no effect. However, looking at the images, it does not appear the same. The authors need to control this. As mentioned before a double mutant analysis would be beneficial to dissect the role of seipin/Ldb16 complex. Its perplexing why would double deletion of nem1 with sei1/Ldb16 would have more LDs. As a matter of fact, WT cells do not show any increase in the number of LDs at 37, show the authors should reason. Under same conditions at 37 (see Fig 2C) the cells do not have more TAG- or SE-containing LDs. Then why deletion of nem1 suddenly only at 37 starts to show this phenotype. This needs to be addresses and explained

9) Fig S1F, looking by eye does not seem much of a difference. Quantification is required, Its not clear if deletion of either Sei1 or Ldb16 have any effect on LRAT induced RE-LDs. Have the authors tested double mutant lacking both sei1 and ldb16 on

LRTAT induced RE-LDs. This result conflicts with recently published paper in JCB (Molennar et al., 2021) It need to be interpreted with caution until you have supporting data. At what temperature this experiment was done is not mentioned. If it was at 30 then what would you expect at 37.

Reviewer #1

1- “The study addresses an important question in the field. However, a major concern is how LD morphology and diameter are being analyzed. It is unclear what statistical approaches are implemented to compare LD diameter among different conditions.”

We have now included a detailed description of the methods used to analyze the LDs in our study. This includes an additional supplemental figure describing the general pipeline used in image processing and quantification as well as the statistical analysis. All the information about statistical significance is now included whenever it is appropriate.

2- “a. In Figure 4B/D for example, there is no indication for statistical analysis with Seipin mutants. It is stated that Seipin SS-AA and SS-DD mutants fail to rescue LD diameter whereas Seipin WT does rescue. The SS-AA and SS-DD look quantitatively very different from one another, so some additional analysis here is required. “

This has now been added.

3- “b. Similarly in Figure 2 B/D, where are no statistics presented, but at 30C it appears that there is no difference in LD diameter in sei1 or ldb16 knockout yeast for SE-only droplets. This should be stated in the text. The 37C experiment does appear to show a clear difference in the representative images, but the LD diameter quantification appears to not show any difference. This should be analyzed with statistics and reconciled.”

Our statistical analysis shows that LDs from Seipin mutant cells are significantly different from the ones observed in WT cells. This is true irrespective of their neutral lipid content and temperature. However, the differences are bigger at 37C as indicated in the text. The lipid changes observed at 37C, in particular the increased levels of SE are likely responsible for the stronger phenotype.

4- “c. A general conclusion is that sei1 and ldb16 are required for the formation of TAG-only or SE-only LDs. However, in several images presented here, it appears that SE-only LDs do not contain the same complement of proteins as TAG-only LDs. In F1, Erg6 does not decorate SE-only LDs like the TAG-only LDs. An alternative model for the study is that SE-only LDs contain a different complement of surface proteins, which affects their stability and morphology. Some discussion is necessary.”

We thank the reviewer for pointing this out. Re-analysis of our imaging data showed that Erg6 localizes equally to WT, TAG-only and SE-only LDs when cells are growing in synthetic complete (SC) medium. We include line intensity analysis showing that Erg6 signal coats LDs stained with BODIPY irrespective of their NL content (Figure S1B). This result was also reported previously by Jacquier *et al.* (J Cell Sci 2011). However, when cells are grown in rich medium (YPD) the enrichment of Erg6 to LDs in SE-only cells is decreased in comparison to WT and TAG-only cells, as noted by the reviewer. This suggests that metabolic differences, potentially due to distinct lipid compositions, affect the targeting of Erg6 to SE-only LDs. Distinct protein complements into TAG and SE LDs have been observed in previous studies in mammalian cells (Hsieh *et al.* – JCell Sci 2012). While interesting, these observations fall beyond the scope of our study that focus on the role of Seipin in the packaging of different neutral lipids. Therefore, since all our analysis of Seipin in the formation of TAG-only and SE-only are made in SC media we removed the localization results in YPD to avoid confusion.

Reviewer #2

1- "The authors claimed that formation of hydrogen bonds between carboxyl esters of neutral lipids and S165/S166 is required for lipid droplet formation.

For instance,

- "The TAG concentrating activity of human Seipin depends on the HH, requiring two serine residues (S165/166) in this helix" on Page 3, line 31.

- "This activity involved the formation of a hydrogen bonds (H-bonds) between conserved hydroxyl-containing residues in the central HH and carboxyl esters present in TAG and SE." on Page 4, line 54.

However, this is not supported by the provided data, and I would soften those conclusions and sentences (perhaps replace S165/S166 with hydrophobic helix?). For instance,

- Distribution of LD diameters in SS-AA (S165 and S166 were mutated to alanine)-containing cells and pSeipin-containing cells are very close in Fig. 4B.

Accordingly, I would suggest updating the following sentence on Page 9, line 201. "Importantly, mutation of Seipin S165/166 either to alanine (SS-AA) or aspartate (SS-DD) failed to restore normal LD morphology in *sei1Δldb16Δ* cells"

- Squalene that does not have any hydrogen donors enriches in the seipin ring (Fig. S2E).

- Fatty acids that do have hydrogen donors do not enrich in the seipin ring (Fig. S2D).

Also, I would suggest that the authors be more cautious about the usage of hydrogen bonds when analyzing the MARTINI simulations. Strictly speaking,

there are no explicit "hydrogen" atoms in MARTINI model coarse-grained simulations. Also, the physical limitations of the MARTINI CG model should be mentioned and cited (see Jarin et al., J. Chem. Theory Comp. 17, 1170-1180 (2021). This model should not be used blindly."

Whilst H-bonds are implicitly present in MARTINI, it is true that they are not explicitly modelled. As such, we have softened our mention of H-bonds in relation to the MD data, and added a section to the discussion discussing these limitations.

In addition, we added a section to the discussion on squalene and FAs as suggested by the referee. We have also rephrased the description of the results regarding mutants of the human Seipin hydrophobic helix.

2- "While I am not an imaging expert, I find the following concerns about Fig. 1A and S1G.

- Figure 1A: Generally speaking, BODIPY 493/503 is known for intercalating with TAG and BODIPY-CE with SE. How do the authors know BODIPY 493/503 is a good SE marker?"

BODIPY 493/503 is an apolar compound and its partitioning into LDs depends on its high lipophilicity, not on specific molecular interactions. As such, BODIPY partitions in the core of LDs of either TAG and SE (shown in Figure 1A, SC -ino condition) and has been used as marker of SE-containing LDs before (for example Gao et al., JCB 2017). In the absence of LDs (*i.e.* no LDs cells) or in cells with very few LDs, BODIPY partitions to lipid bilayers labelling virtually all cellular membranes.

In contrast, BODIPY-CE consist of cholesterol esterified to a BODIPY-containing fatty acid. BODIPY-CE is mainly used for tracking CE transport (which specificity depends on the cholesteryl moiety), but due to its apolar chemical nature can also be used to stain LDs, irrespective of their neutral lipid content.

3- "- Figure 1A-YPD-SE-only-BODIPY: Why is there a ring-like intensity in the leftmost lipid droplet? You might need to zoom in this figure to see this. Is there a phase separation as shown in Julia Mahamid et al., PNAS, 2019, 116 (34)? See Fig S3A of this paper. "

In SE-only cells grown in YPD there are very few LDs and as mentioned above, the BODIPY labeling is coming primarily from membranes. This membrane labeling likely explain the ring-like structures observed. However, we have not performed ultrastructural characterization of these cells or have any additional evidence for SE phase separation in these LDs. In addition, a recent preprint (Rogers et al.;

doi.org/10.1101/2021.08.30.458229) shows that SE phase transition in LDs requires TAG lipolysis, which does not occur in these cells.

4- “Erg6 seems to target TAG-LD but does not target SE-LD in Fig. 1A. Given this concern, how do the authors argue that Erg6 is a good LD marker for retinyl esters LD in Fig. S1G? Also, considering this, Erg6 should be referred to as the TAG-LD marker protein rather than simply the LD marker protein.”

As mentioned above (Reviewer 1, Point 4), we re-analyzed Erg6 localization in SE-only cells. We observe that Erg6 localizes efficiently to SE-only LDs in cells grown in SC media while its targeting is decreased to SE-only LDs from cells grown in YPD. In fact, for cells grown in SC media, Erg6 localizes to all LDs irrespective of their neutral lipid content (and including RE LDs). To avoid any confusion, we removed the experiment performed in YPD (initially shown in Figure 1A and S1C). In addition, line intensity analysis further show that Erg6 signal coats LDs stained with BODIPY irrespective of their NL content (Figure S1B).

5- “The authors made new molecules (SE, RE, and SQL) in the MARTINI simulations. Have the authors created “new” atomic types that are not included in the original release of MARTINI? If yes, please indicate all the details. Otherwise, the atomic types used should be clearly indicated in Methods and Fig 3.”

Only the SQL parameters are new to this study, and these used standard MARTINI bead types. We have made all the details, including bead types, freely available on the OSF (<https://osf.io/wnzsf/>). We have also added a section describing the atomistic simulations and how these were used to parameterize the new lipid.

6- “The authors calculated the following quantities in their MARTINI simulations: lipid enrichment, lipid occupancy, and binding duration. However, how they characterized those quantities are not described.”

We have now clarified in the text and methods sections how each of the relevant lipid analyses were run.

7- “Please indicate atomic types of TAG, RE, and SQL. Regarding atomic types, I have the following concerns:

- Do the following atoms have the same atomic types: C3A, C3B, C3C, and C3D of TAG and C3, C4, C5, C6 of SE? Are C3B and C3C of TAG the same atomic types? If so, the MARTINI force fields that the authors used seem not to consider the

double bonds in the acyl chains. Therefore, TAG represents tripalmitin (solid at room temperature), not triolein (the most abundant fat).

- Although there is no direct mapping scheme between all-atom and MARTINI CG force fields, the authors should clearly describe what atomic types were used for the glycerol moieties. For instance, what is GLY of TAG in Fig. 3E? Please note that such a grouping (GLY) does not exist in phospholipids in MARTINI force fields."

For this manuscript, we have only parameterized SQL, with the CE, RE and TAG parameterized elsewhere (and referenced in our manuscript). We have provided the bead types along with the parameters at <https://osf.io/wnzsf/>.

The TAG has previously been described in Vuorela et al PLOS CompBio 2010, including all of the bead types. It is our understanding that these have been accurately modelled, including inclusion of the double bonds in the lipid tail. We have clarified that in our manuscript the bead *names* used do not reflect the actual bead *types*.

8- "As SE, RE, and SQL are new molecules (and TAG also, as it was not constructed in a standard way, e.g., inclusion of GLY that is not included in phospholipids and lack of double bonds in acyl chains), the authors should characterize the physical properties of those molecules and compare those with experimental data, e.g., phase at room temperature, density, interfacial tension against water, lattice parameters if solid at room temperature."

As stated above, only the parameters for SQL are new in our manuscript. These were parametrized against atomistic simulation and experimental LogP values. We have expanded the description of both in the methods section.

9- "Coarse-grained simulations have advantages of exploring larger length scales and longer time scales. However, the MARTINI simulations here only showed "recruitment" of neutral lipids but no "nucleation" of neutral lipids. Do neutral lipids eventually nucleate? Given the system sizes shown here, I am certain that the authors can extend the simulations. Please note that all the other computational simulations about seipin showed "nucleation"."

This is an interesting point. In our model, Seipin recruits and enriches NLs. This increased local concentration will enhance NL-NL interactions, and drive NL nucleation. Our study focusses on the initial stages of NL-concentration by Seipin, specifically the direct Seipin-NL interactions. Thus, using low concentrations of NL, we show that Seipin can enrich NLs in its ring-like structure, with NLs positioning close to

the HH. When we have run much longer simulations, no additional NL enrichment was observed.

Although some MD-studies have reported data on NL nucleation, many others (such as Zoni *et al.* (PNAS 2021), Prasanna *et al.* (PLoS Biol. 2021) and (Klug *et al.* – Nat Comm 2021)) also show mainly enrichment of TAG molecules to Seipin. This makes us confident that the phenomena that we are reporting here relate to Seipin function.

10- “As a control experiment, simulations of pure lipid systems (no seipin) should be shown as well as to whether neutral lipids nucleate at critical concentrations.”

The requested simulations have been run and are now included in the manuscript (Figure S3H).

11- “Figure S2: There is no system that has both DAG and TAG according to Table S3. Either Figure S2 or Table S3 should be updated.”

The information in Table S3 has been corrected.

12- “Page 4, line 41: “In contrast, foam cells, lipid-laden macrophages found in atherosclerotic lesions, contain LDs that mainly consist of SE”. Add references.”

Reference has been added.

13- “Figure 1A and Page 5, line 65: Do “quadruple mutant (QM)” and “no NLs” represent the same cells (dga1Δro1Δare1Δare2Δ)? For consistency, I would use either of them.”

Done.

14- “Figure 2E right: I would add the x label in the plot although it is explained in the legend. “

Done.

15- “Page 7 line 140: Acronym RE was already mentioned in the Introduction. “

This has been fixed.

16- “Page 8, line 189: Different insertion depths of PL and TAG carboxyl esters were already shown in bilayers [Campomanes et al., Biophysical Reports, 2021, 1(2); Kim and Voth, JPCB, 2021, 125(25)]. Cite those papers.”

The references have been added.

17- “Figure S2: For consistency, use either SE or CE. “

Done.

18- “Page 11, line 258: “Both NLs and PLs contain carboxyl ester groups that could in principle accept H-bond from Seipin hydroxyl residues. However, the projection of Seipin HH deep into the membrane bilayer, positions S165/166 close to the NL carboxyl esters at the center of the bilayer and distant from the PL carboxyl esters, which are at the membrane-water interface”. The same perspective was discussed in Kim et al., bioRxiv 10.1101/2021.12.05.471300. Perhaps, cite this paper? “

Done.

19- “Page 11, line 267: Acronym CCT is not introduced before. Spell it out.”

Done.

20- “Page 14, line 363: What is a flat-bottomed distance restraint? Is it another elastic network? “

A flat bottomed distance restraint is a distance restraint between two reference groups in three-dimensional space, which takes the form of a flat-bottomed potential rather than a harmonic potential, as is more standard. This allows for great flexibility of ranges without incurring penalty forces. We have clarified this in the text.

21- “Page 14, line 368: In the initial structures, how many phospholipids were included inside the hydrophobic ring of seipin in the luminal leaflet? “

Lipids were placed automatically by the insane.py script. Approximately 16 lipids were in the luminal leaflet, in the pore formed by the central Seipin helix.

22- “Please include page numbers in the manuscript.”

Done.

Reviewer #3

1- “Figure 1A, why SE-only cells are making less LDs. The authors need to explain this. Is TAG required for nucleation of SE-only LDs? In the same figure see panel SE-only, why Erg6 is not in localizing to few BODIPY puncta that represent SE-LDs. It's not clear the way it is presented.”

We initially suspected that SE-only cells have less LDs due to low levels of NLs. Our new lipidomics analysis supports this idea by showing that SE-only cells have roughly half of the amount of NLs of WT and TAG-only cells, when grown at 37C and even less when grown at 30C (see new Fig. 2). We have also made changes in Figure 1 and in the text to clarify the localization of Erg6 in SE-only cells (See Reviewer 1, point 4).

2- “In Fig 1A, panel no NLs, I see a lot of fluorescence of BODIPY in the ER that colocalizes with Erg6. Does this mean the cells have NLs that is dispersed in the ER. Apparently, the cells should have no NLs at all. It needs to be controlled with lipid analysis in this mutant in both culture conditions. As a matter of control see image SE-only, I don't see fluorescence of BODIPY in the ER in cells that do not have any LDs. “

BODIPY is a lipophilic molecule with great avidity for NLs. However, in absence of NLs (for example in no NLs cells), BODIPY partitions and labels other hydrophobic structures such as cellular membranes. This is a well-established observation and widely reported in the literature. We have also change the text to make this clear.

3- “Fig 1C and D: Since sei1 and ldb16 are part of a complex, it would be advantageous to detertmine what % of Sei1 and ldb16 colocalize in cells containing only TAG-LDs, SE-LDs or no-NLs. Fig 1E, western blot to determine if Pln1 levels are downregulated in cells containing no NLs and compare to other conditions. “

The study by Wang et al, JCS (2014) showed that the majority of Sei1 and Ldb16 foci in cells colocalize. This is consistent with the observation that Ldb16 is unstable when not bound to Sei1. In contrast, Sei1 stability is not affected by Ldb16.

As suggested by the reviewer we have now analyzed the levels of Pln1 in cells with the various genotypes by immunoblotting (Figure S1D). As expected, Pln1-levels are similar in WT and TAG-only cells. In SE-only cells Pln1-levels are reduced compared to WT, but still observable, whereas the mutant with no NLs has nearly no detectable

Pln1. Notably, when SE-only cells were cultured at 37°C to stimulate the formation of SE-only LDs, Pln1-levels are higher when compared to SE-only cells cultured at 30°C. This result suggests that available LD surface is important in determining Pln1 levels irrespective of NL type, and is consistent with our conclusions.

4- “The authors claim that since SE-only cells have very few LDs, hence it complicated their analysis of LD morphology. However, surprisingly, at 37 they see more SE containing LDs while TAG-containing LDs did not show any change whether grown at 30 or 37. I don't understand this at all. This is an important point. Many experiments are based on this observation. So first the authors should perform lipid analysis to quantitatively determine if SE levels are in-fact higher when cells are shifted to 37, which they claim that the cells are making high SE at 37. The authors should also reason why SE would be higher at 37 and not TAG. Are the two SE-acyltransferases expressed at higher levels at 37. “

This is an excellent point and we thank the reviewer for suggesting the lipidomics analysis. The lipid measurements performed by mass spectrometry explain the temperature effects on LDs. Consistent with our imaging data, we observe very minor differences in the levels of NLs in WT and TAG-only cells. In contrast, there is a measurable increase in the amounts of SE at 37°C in comparison to cells grown at 30°C. Moreover, we observe that overall content of NLs is higher in WT and TAG-only cells consistent with increased numbers of LDs in these cells in comparison to SE-only. Thus, the lipidomics data match our imaging analysis.

The temperature-dependent increase in SEs is interesting but the reasons behind it fall outside the scope of this study. As we mention the temperature dependent increase in SE-only LDs came from a fortuitous observation that we exploited to study the role of Seipin in SE packaging into LDs.

5- “I have several issues with Figure 2, the way authors have interpreted the images and how they have presented. It is well known that cells lacking either Sei1 or Ldb16 have supersized LDs. In Fig 2A-D, I don't see accumulation of supersized LDs in cells containing only SE-containing LDs, though you can clearly see in TAG-containing LDs. Why is it so. Even in the quantification for LD diameter, there is hardly any difference for Wt, sei1, and ldb16 mutants in SE-only LDs. This is central part of the paper. This need to be controlled properly.”

Work from many groups, including ours, showed that cells lacking Sei1, Ldb16 or both have defects in LD morphology displaying populations of both tiny and supersized LDs. Thus, in contrast to WT cells, in which the size of LDs follows a Gaussian (or normal) distribution, the size of LDs in Seipin mutants follow a skewed non-normal

distribution. As shown in Figures 3B and D, these non-normal distribution of LD size in Seipin mutants was observed irrespective of the NL content and temperature. Importantly, statistical analysis shows that Seipin mutations display significantly different LD size distributions in relation to WT cells. We changed to text to clarify this point. Information on image acquisition, processing and analysis as well as data quantification are now described in further detail (Figure S1 and material and methods section).

6- “It will be interesting to add if both *sei1* and *ldb16* are missing, what would happen to SE-only and TAG-only LDs at 30 and 37. It is important to know if the effect is synergistic. More data are needed to define how it occurs.”

We have now included the analysis of *sei1Δldb16Δ* cells. As expected, these cells behave like the individual mutants. In fact, given that *Ldb16* is unstable in the absence of *Sei1*, *sei1Δ* and *sei1Δldb16Δ* are largely the same.

7- “In S1D, I see clear pattern of the role of *Sei1/ldb16* in LD defects in TAG-containing LDs when cells are stimulated to make de novo LDs. However, comparing the same induction for SE-only LDs, the role of does not seem that apparent. Cells have very few LDs, I don't see clustered or supersized LDs, a classical phenotype of seipin mutants. The majority of cells seem to form at least 1 LD upon 180 min of induction. So it's difficult to attribute a clear role of *Sei1* in the biogenesis of SE-only LDs. The authors should perform lipid analysis to see if there is any difference. Similarly, the authors should perform induction with *ARE2* to see if it would have dramatic effect of LD morphology defects. “

As pointed out by the reviewer, induction of the TAG acyl transferase *Dga1* in No LD cells lacking Seipin results in dramatic defects in LD biogenesis. This phenotype has been reported by us and other before. Now we show that, in the same cells, induction of the SE acyltransferase *Are1* results in LD biogenesis demonstrating a role for Seipin in SE-only LD biogenesis. In figure 3E, we observe that the kinetics of LD formation upon *ARE1* induction is much slower in Seipin mutants in comparison to control cells. Even at later timepoints (180min), we observe much lower number of LDs being formed in Seipin mutant cells. In addition, aberrant LD morphology is also observed upon prolonged expression (24h) of *Are1* or *Are2* in the same cells (Figure S2D).

8- “Quantification for LD size in S1F is required. Here the authors need to perform lipid analysis to see if at 37 when cells are lacking *nem1* together with *sei1/ldb16* are now making more TAG. Earlier they mentioned that at 37 only SE-containing LDs are more and TAG-LDs have no effect. However, looking at the images, it does not appear the same. The authors need to control this. As mentioned before a double mutant analysis would be beneficial to dissect the

role of seipin/lhb16 complex. Its perplexing why would double deletion of nem1 with sei1/lhb16 would have more LDs. As a matter of fact, WT cells do not show any increase in the number of LDs at 37, show the authors should reason. Under same conditions at 37 (see Fig 2C) the cells do not have more TAG- or SE-containing LDs. Then why deletion of nem1 suddenly only at 37 starts to show this phenotype. This needs to be addressed and explained

We quantified the size of LDs in cells deleted for Nem1, as requested. We observe that while in *nem1Δ* cells the LD size follows a Gaussian distribution, loss of Sei1, Lhb16 or both in the *nem1Δ* mutant background results in skewed non-normal distributions. As described before, the non-normal distribution reflects the presence of 2 populations of LDs in the Seipin mutant backgrounds, with LD being either very small or supersized. The phenotype of Seipin deletion in the *nem1Δ* background can be appreciated in the graphs with the individual LD size measurements and statistical analysis shows that it is significantly different from single *nem1* deletion. The data are presented in Figure S2C.

9- "Fig S1F, looking by eye does not seem much of a difference. Quantification is required, Its not clear if deletion of either Sei1 or Lhb16 have any effect on LRAT induced RE-LDs. Have the authors tested double mutant lacking both sei1 and lhb16 on LRAT induced RE-LDs. This result conflicts with recently published paper in JCB (Molenaar et al., 2021) It need to be interpreted with caution until you have supporting data. At what temperature this experiment was done is not mentioned. If it was at 30 then what would you expect at 37.

We quantified the size RE LDs, as requested. Sei1 and Lhb16 mutant cells showed an increase of supersized LDs. In addition, Seipin mutations also resulted in higher number of tiny LDs. These data are shown in Figures S2F and G. Quantification of RE LDs was performed differently because for unknown reasons these LDs were not efficiently stained by BODIPY. These experiments were performed at 30C and this information is now included in the figure legend.

July 5, 2022

RE: JCB Manuscript #202112068R

Dr. Pedro Carvalho
University of Oxford
Sir William Dunn School of Pathology
South Parks road
South Parks Road
Oxford, UK OX1 3RE
United Kingdom

Dear Dr. Carvalho:

Thank you for submitting your revised manuscript entitled "Seipin concentrates distinct neutral lipids via interactions with their acyl chain carboxyl esters". As you will see, Reviewers 1 and 2 are fully supportive of publication of the current study. We agree the requested discussion points of Reviewer 1 need to be included. Reviewer 3 still has questions regarding the underlying mechanism, in particular pertaining to the effects of temperature. While we appreciate these points, we find that you have addressed all essential concerns from the first set of reviews, and that further insight for example into expression of SE-acyltransferases can be examined in follow up studies. Therefore, we would be happy to publish your paper in JCB pending final revisions necessary to meet our formatting guidelines (see details below).

A. MANUSCRIPT ORGANIZATION AND FORMATTING:

- 1) Text limits: Character count for Articles is < 40,000, not including spaces. Count includes abstract, introduction, results, discussion, and acknowledgments. Count does not include title page, figure legends, materials and methods, references, tables, or supplemental legends.
- 2) Figures limits: Articles may have up to 10 main text figures.
- 3) Figure formatting: Scale bars must be present on all microscopy images, including inset magnifications. Molecular weight or nucleic acid size markers must be included on all gel electrophoresis.
- 4) Statistical analysis: Error bars on graphic representations of numerical data must be clearly described in the figure legend. The number of independent data points (n) represented in a graph must be indicated in the legend. Statistical methods should be explained in full in the materials and methods. For figures presenting pooled data the statistical measure should be defined in the figure legends. Please also be sure to indicate the statistical tests used in each of your experiments (either in the figure legend itself or in a separate methods section) as well as the parameters of the test (for example, if you ran a t-test, please indicate if it was one- or two-sided, etc.). Also, if you used parametric tests, please indicate if the data distribution was tested for normality (and if so, how). If not, you must state something to the effect that "Data distribution was assumed to be normal but this was not formally tested."
- 5) Abstract and title: The abstract should be no longer than 160 words and should communicate the significance of the paper for a general audience. The title should be less than 100 characters including spaces. Make the title concise but accessible to a general readership.
- 6) Materials and methods: Should be comprehensive and not simply reference a previous publication for details on how an experiment was performed. Please provide full descriptions in the text for readers who may not have access to referenced manuscripts.
- 7) * Please be sure to provide the sequences for all of your primers/oligos and RNAi constructs in the materials and methods. You must also indicate in the methods the source, species, and catalog numbers (where appropriate) for all of your antibodies. Please also indicate the acquisition and quantification methods for immunoblotting/western blots. *
- 8) Microscope image acquisition: The following information must be provided about the acquisition and processing of images:
 - a. Make and model of microscope

- b. Type, magnification, and numerical aperture of the objective lenses
- c. Temperature
- d. Imaging medium
- e. Fluorochromes
- f. Camera make and model
- g. Acquisition software
- h. Any software used for image processing subsequent to data acquisition. Please include details and types of operations involved (e.g., type of deconvolution, 3D reconstitutions, surface or volume rendering, gamma adjustments, etc.).

10) Supplemental materials: There are strict limits on the allowable amount of supplemental data. Articles may have up to 5 supplemental figures. Please also note that tables, like figures, should be provided as individual, editable files. A summary of all supplemental material should appear at the end of the Materials and methods section.

13) ORCID IDs: ORCID IDs are unique identifiers allowing researchers to create a record of their various scholarly contributions in a single place. At resubmission of your final files, please consider providing an ORCID ID for as many contributing authors as possible.

Please note that JCB now requires authors to submit Source Data used to generate figures containing gels and Western blots with all revised manuscripts. This Source Data consists of fully uncropped and unprocessed images for each gel/blot displayed in the main and supplemental figures. Since your paper includes cropped gel and/or blot images, please be sure to provide one Source Data file for each figure that contains gels and/or blots along with your revised manuscript files. File names for Source Data figures should be alphanumeric without any spaces or special characters (i.e., SourceDataF#, where F# refers to the associated main figure number or SourceDataFS# for those associated with Supplementary figures). The lanes of the gels/blots should be labeled as they are in the associated figure, the place where cropping was applied should be marked (with a box), and molecular weight/size standards should be labeled wherever possible.

B. FINAL FILES:

**It is JCB policy that if requested, original data images must be made available to the editors. Failure to provide original images upon request will result in unavoidable delays in publication. Please ensure that you have access to all original data images prior

to final submission.**

Thank you for this interesting contribution, we look forward to publishing your paper in Journal of Cell Biology.

Sincerely,

Jodi Nunnari, Ph.D.
Editor-in-Chief

Andrea L. Marat, Ph.D.
Senior Scientific Editor

Journal of Cell Biology

Reviewer #1 (Comments to the Authors (Required)):

The revised manuscript has addressed my major issues. More robust statistical analysis and quantification has been employed. The study utilizes cell and modeling experiments to conclude that seipin contributes to the coalescence of several neutral lipids in the ER bilayer.

A minor concern remains: in most cell types, combinations of these neutral lipids will exist in the ER at the same time. An expectation is that these neutral lipids will influence one another's coalescence and loading into droplets, which may also influence how droplets form, as well as how seipin interacts with each neutral lipid species. Although beyond the scope of this study, some more comments in the Discussion on how different neutral lipids influence each other during LD biogenesis seems appropriate. Also adding that a limitation of this study is using primarily strains that contain TAG or SE only should be discussed, as these conditions differ from typical cellular neutral lipid profiles.

Reviewer #2 (Comments to the Authors (Required)):

I'm happy with the revision.

Reviewer #3 (Comments to the Authors (Required)):

The authors have worked on the constructive feedback and have improved the manuscript to some extent. Some of the concerns raised upon initial submission have been addressed in the revised version. However, the study lacks a mechanistic understanding. Though the observations are intriguing, the findings are descriptive. Some of the key findings have not been clearly spelled out. Importantly, the temperature dependent changes in the lipid profile of TAG-only and SE-only cells were reported (Fig. 2). There is a tiny increase in LD diameter of SE-only cells when they were shifted to 37 degrees that correlates with lipidomic analysis of elevated SE levels (Fig 2C, green bars). But on the other hand, TAG-only cells despite showing significant increase in TAG levels by lipidomic analysis (Fig 2C, pink bars), resulted in a decreased LD diameter (Fig. 2B). What could be the reason for this. Are the two SE-acyltransferases expressed at higher levels at 37 degrees. I had asked to determine this. It could have provided some hint in the mechanism of LD assembly. The authors have not explained or followed such

observations. Therefore, the data appears preliminary with no mechanistic insight. Loss of Seipin or Ldb16 results in heterogeneous LD population, it has been known from previous study from the same group and others. A delayed kinetics of LD formation at 30 degrees upon induction of SE-only in cells devoid of Seipin function is similar to what others have observed for TAG-only cells. Comparison of Induction at 37 degrees is missing to assess if elevated SE levels as per lipidomic measurement, would be able to rescue delayed induction in cells lacking Seipin function. It would have uncovered some mechanistic view of what is going on at 37. Although there are some interesting observations, the current study does not add considerably to the mechanistic understanding of LD biogenesis how it differs at different temperatures as the study is focused on.

Dear Jodi and Andrea,

We were thrilled to hear that JCB is ready to move forward with the publication of our study.

We are now submitting a revised manuscript in which we addressed the minor issue raised by reviewer #1, as requested. In addition, we took care of all formatting issues there were still pending.

We hope that with these changes the paper will now be acceptable for publication in JCB.

Sincerely,

Pedro

Response to Reviewer #1

“The revised manuscript has addressed my major issues. More robust statistical analysis and quantification has been employed. The study utilizes cell and modeling experiments to conclude that seipin contributes to the coalescence of several neutral lipids in the ER bilayer.

A minor concern remains: in most cell types, combinations of these neutral lipids will exist in the ER at the same time. An expectation is that these neutral lipids will influence one another's coalescence and loading into droplets, which may also influence how droplets form, as well as how seipin interacts with each neutral lipid species. Although beyond the scope of this study, some more comments in the Discussion on how different neutral lipids influence each other during LD biogenesis seems appropriate. Also adding that a limitation of this study is using primarily strains that contain TAG or SE only should be discussed, as these conditions differ from typical cellular neutral lipid profiles”

We agree with the reviewer that most cells contain more than a single neutral lipid type in the core of the lipid droplets. We have added a new paragraph to the discussion section where this limitation is mentioned. This paragraph also points out that there are a few exceptional cases in which the core of lipid droplets consists almost exclusively of a single neutral lipid. These are the cases of adipocytes, foam cells and hepatic stellate cells whose lipid droplets are filled respectively by TAG, SE and RE.